# Geographically Disperse, Culturable Seed-Associated Microbiota in Forage Plants of Alfalfa (*Medicago sativa* L.) and Pitch Clover (*Bituminaria bituminosa* L.): Characterization of Beneficial Inherited Strains as Plant Stress-Tolerance Enhancers

**DOI:** 10.3390/biology11121838

**Published:** 2022-12-16

**Authors:** Marla Niza-Costa, Ana Sofía Rodríguez-dos Santos, Inês Rebelo-Romão, María Victoria Ferrer, Cristina Sequero López, Juan Ignacio Vílchez

**Affiliations:** 1iPlantMicro Lab, Instituto de Tecnologia Química e Biológica (ITQB)-NOVA, Oeiras, 2784-501 Lisboa, Portugal; 2GeoBioTec, Department of Earth Sciences, NOVA School of Sciences and Technology, Universidade NOVA de Lisboa (Campus de Caparica), 1070-312 Caparica, Portugal

**Keywords:** *Medicago sativa*, *Bituminaria bituminosa*, seedborne microbiota, drought treatment, beneficial microbiota inheritance, *Paenibacillus*, *Stenotrophomonas*

## Abstract

**Simple Summary:**

The implementation of new strategies to define beneficial bacteria for improving tolerance to stressful conditions in plants is becoming a very relevant topic. The use of microbiota associated with seeds may generate a new solution to improving the efficiency of bioinoculants. Legume production suffers from increasingly accentuated climatic conditions such as heat and drought periods. This group of plants can be especially protected against stress by applying strains isolated from seeds of wild legume varieties. This study shows how strains isolated from seeds of alfalfa and pitch clover are capable of improving the development of lentil plants under drought conditions.

**Abstract:**

Agricultural production is being affected by increasingly harsh conditions caused by climate change. The vast majority of crops suffer growth and yield declines due to a lack of water or intense heat. Hence, commercial legume crops suffer intense losses of production (20–80%). This situation is even more noticeable in plants used as fodder for animals, such as alfalfa and pitch trefoil, since their productivity is linked not only to the number of seeds produced, but also to the vegetative growth of the plant itself. Thus, we decided to study the microbiota associated with their seeds in different locations on the Iberian Peninsula, with the aim of identifying culturable bacteria strains that have adapted to harsh environments and that can be used as biotreatments to improve plant growth and resistance to stress. As potentially inherited microbiota, they may also represent a treatment with medium- and long-term adaptative effects. Hence, isolated strains showed no clear relationship with their geographical sampling location, but had about 50% internal similarity with their model plants. Moreover, out of the 51 strains isolated, about 80% were capable of producing biofilms; around 50% produced mid/high concentrations of auxins and grew notably in ACC medium; only 15% were characterized as xerotolerant, while more than 75% were able to sporulate; and finally, 65% produced siderophores and more than 40% produced compounds to solubilize phosphates. Thus, *Paenibacillus amylolyticus* BB B2-A, *Paenibacillus xylanexedens* MS M1-C, *Paenibacillus pabuli* BB Oeiras A, *Stenotrophomonas maltophilia* MS M1-B and *Enterobacter hormaechei* BB B2-C strains were tested as plant bioinoculants in lentil plants (*Lens culinaris* Medik.), showing promising results as future treatments to improve plant growth under stressful conditions.

## 1. Introduction

In recent decades, we have been enduring the acceleration of climate change’s effects. This situation has caused a progressive impact on many crops worldwide, which will severely affect agricultural production and food supply in oncoming years [1,2,3]. In the case of the Mediterranean basin and Southern European countries, the main manifestation of these effects is the increase in the duration and intensity of heat waves and of drought periods [4,5]. Moreover, these conditions are also aggravating other threatening situations in these areas, affecting factors such as salinity, nutrient availability and the spread of pests and pathogens [6,7]. Thus, during spring, summer and autumn, that is, the main crop seasons, these climatic conditions are exacerbated, bringing poor prospects for agriculture and crop management.

Due to their nutritional richness and their relevance in the diets of these countries (and an increasing number of countries), legumes have become one of the pillars to guarantee food security [8,9]. In addition, these crops are also some of the main foods to guarantee livestock production [10]. However, the commercial species that are usually harvested (lentils, chickpeas, beans, broad beans and peas) are generally sensitive to a lack of water and heat waves [11,12]. In addition to the effects of water stress on plants, under harshening conditions, the symbiosis that these crops produce with soil microorganisms is attenuated or even prevented, causing worse overall development and a notable loss of productivity [12,13,14].

Despite this, drought events do not have the same impact on foraging legumes or other legumes adapted to arid environments [15,16]. It is probable that the reduced or lack of human domestication of these plants has preserved the characteristics of resistance to stress in these species [17,18,19]. This does not only imply a more adaptative genetic background, but also suggests better communication and interaction with beneficial soil microbiota [20]. Recently, researchers have described, in modern crop species, how domestication has affected the associated microbiota and the types of interactions that plants are capable of carrying out [21,22,23]. Hence, the selection of more desirable characteristics, such as the number, size or coloration of fruits, has conditioned the persistence of the mechanisms of adaptations and responses to stress [24,25]. Among them, the ability to attract and interact with beneficial microorganisms through changes in the composition of secretions (root exudates, volatile organic compounds and foliar waxes) is one of the most relevant [26,27]. This is not only because these interactions improve the response and resistance of plants, but they also affect signaling mechanisms that enable microbiota inheritance through the seeds [28,29].

In spite of this scenario, in recent decades, some alternatives have been developed to improve this situation. The use of beneficial bacteria as biotreatments is on the rise, since they are a sustainable and ecologically friendly alternative [30,31,32]. Among other mechanisms, these bacteria improve the response to different stresses by protecting the roots (exopolysaccharides and biofilms) and improving their number, length and distribution through the production of auxins, thereby reaching less accessible water sources. In addition, they are capable of regulating other phytohormones such as ethylene through the production of ACC (1-aminocyclopropane-1-carboxylate) deaminase [30,33]. The action of this enzyme prevents the accumulation of ethylene during periods of drought, thus reducing the loss of leaves and roots, which allows more complex root architecture to be maintained [34]. Other desirable characteristics of these strains are the increase in nutrient accessibility, biocontrol skills and xero/osmoprotectant compound production [33,34,35].

Thus far, the most common sources of drought-tolerant enhancer bacteria have been desert soils or the rhizosphere of adapted plants [30,36,37], but new emerging isolation sources could be even more suitable. In this sense, the microbiota associated with seeds may represent a great advancement in these types of treatments [38]. Despite their variability and the possibility of isolating pathogens that have been able to surpass plants’ barriers, seed microbiota are provided as support from mother plants to their future offspring [39,40,41,42,43]. Hence, these microbiota usually include beneficial populations that must be better adapted to carrying out interaction processes and that are eventually selected to overcome stressful events suffered by the previous generation [44,45,46]. There is not much information in this field yet, but the prospects are very high because recent advances in metagenomics seem to indicate that these assumptions are correct [47,48].

With this in consideration, the isolation and characterization of the microbiota associated with seeds in plants adapted to dry environments has begun to generate much interest. In the case of legumes, forage plants such as alfalfa (*Medicago sativa* L.) or pitch clover (*Bituminaria bituminosa* (L.) C.H.Stirt) are some of the best options for obtaining new beneficial strains [49,50]. Both species are found either in crops for foraging or naturally in desert-like environments of the Iberian Peninsula. In addition, in both plants, beneficial associations have been described in nodules and roots. In the specific case of alfalfa, it is also a crop with lower handling requirements and higher nutritional values than other generally used forage crops (corn, soy and sugar cane) [51]. On the other hand, pitch clover, apart from having characteristics similar to those mentioned for alfalfa, is known for being able to maintain its ability to fix nitrogen under drought conditions, which indicates its capability to maintain microbial interactions during these stressful events [52]. Likewise, both plants survive the harsh conditions of the Mediterranean summer and produce their seeds at the same time. As both of these plants are used as animal fodder, they have a problem with not only seed production but also plant development, since the entire plant is used to feed animals and their productivity also depends on their vegetative growth [53,54].

In this context, our work seeks to isolate, identify and characterize the microbiota of alfalfa and pitch clover seeds from different arid and semi-arid regions of the Iberian Peninsula, in order to use them as biotreatments against drought in legume crops.

## 2. Materials and Methods

### 2.1. Sampling Process

The sampling process was focused on wild alfalfa (*Medicago sativa* L.) and pitch clover (*Bituminaria bituminosa* (L.) C.H.Stirt.) as model plants. This process was carried out during July and August 2022, in 9 locations for each type of plant, as indicated both in Table 1 and Figure 1. The sampling locations were selected based on the presence of the model plants in the geographical regions defined as arid and at high risk of desertification by the European Union [55,56]. We included the location of ‘Oeiras’ (Lisbon, Portugal) as a control point as it is a geographically and climatically different location compared to the others proposed in this study. This control point has been identified as a humid/sub-humid area, with no risk of desertification [55]. Once adequately identified, a soil sample was collected close to their roots, at a depth >10 cm. They were characterized based on their pH (as in Mclean, 1983 [57], with minor modifications), their salinity (electroconductivity (EC), measured by using an electrical conductivity meter (Crison, Spain)) and their percentage of water content (as in Li & Wang, 2014 [58]; weight difference after 7 days of incubation at 60 °C). In addition, between 30 and 100 pods in each location were sampled. The seeds were cleaned of pod and plant remains. Both samples were kept at 4 °C until processing.

### 2.2. Germination Process

To carry out cleaner germination, a 60-seed set from each location was surface-sterilized via immersion in 70% ethanol for 5 min under agitation, followed by 10 min in 20% bleach solution. After 3 washes with sterile double-distilled water, the seeds were incubated in water overnight at room temperature in order to improve their hydration and germination rates. After incubation, the seeds were placed on a wet paper towel inside a Magenta box (Biogen Científica, Madrid, Spain), and incubated in dark conditions. The paper was regularly moistened and replaced if an odor or microbial growth appeared. This same procedure was carried out for lentil seeds. After 7 days of incubation, the germination rate (%) was calculated. This experiment was repeated 3 times.

### 2.3. Culturable Bacteria Isolation, Identification and Collection Storage

Following the previous procedure, the roots of the germinated seedlings were surface-sterilized to isolate only endophytic or colonizing strains (epiphytic strains capable of creating resistant biofilms), while also avoiding any other external contamination [60]. The population for each location was calculated based on the number of colony-forming units (CFUs) per milligram of root dry weight (DW). Morphologically different colonies (as in the American Society for Microbiology; [61]) were selected, isolated and purified on fresh plates of LB Agar (per liter: NaCl, 10 g; yeast extract, 5 g; tryptone, 10 g; agar, 15 g). Once the pure cultures were assessed, they were considered the collection strains for this work and were preserved in 40% glycerol as stock.

In order to identify these strains, genomic DNA was extracted from the isolated colonies using the heat shock method, followed by centrifugation for 5 min at 14,000 rpm to separate DNA from the bacteria remains [62]. The concentration and quality of the extracted DNA were checked through Nanodrop spectroscopy (Thermo Scientific™ NanoDrop™ One Microvolume UV-Vis Spectrophotometer, Thermo Scientific™, Waltham, MA, USA). Then, the hypervariable V5–V8 region of the 16S rRNA gene (about 700 bp) was amplified via PCR using the universal primers 779F (5′-AACMGGATTAGATACCCKG-3′) and 1392R (5′-GGTTACCTTGTTACGACTT-3′), and NZYTaq II 2× Master Mix by Nzytech, Lisbon, Portugal. The amplification conditions were: an initial denaturation phase (2 min) at 95 °C; 40 cycles of denaturation (3 s) at 95 °C, annealing (30 s) at 45 °C, and elongation (2 min) at 72 °C; and a final extension phase (7 min) at 72 °C. The integrity of the amplicon was assessed through an electrophoresis gel (1% agarose) using a Mupid^®^-exU System (Advance Co., Ltd., Bangkok, Thailand) at 135 V, and thereafter, a transilluminator (UVIvue™; Daihan Science, Seoul, Republic of Korea). Finally, the samples were sent to GENEWIZ (Leipzig, Germany) for sequencing, and the resulting sequences were compared to the National Library of Medicine BLAST database to identify the strains (>98% similarity). Strains were considered ‘Unidentified’ when we were not able to amplify the 16S rRNA or when their quality was not high enough for sequencing after several attempts.

#### Phylogenetical Trees and Venn Diagrams: Analysis of Culturable Populations

As an identification assessment and in order to evaluate the isolated strains’ genetic distances, phylogenetical trees were prepared. For this, the V5–V8 amplicon sequences were aligned in ClustalX2 (v2.0; University College Dublin, Dublin, Ireland), and the trees were visualized using the iTol drawing tool (https://itol.embl.de/itol.cgi) [63]. To determine representative overlapping of strains among different regions and plant species, the Venn Diagram tools from VIB/UGent—Bioinformatics & Evolutionary Genomics (Gent, Belgium) (http://bioinformatics.psb.ugent.be/webtools/Venn/) and from InteractiVenn [64] were employed (accessed on 1 September 2022).

### 2.4. Plant-Growth-Promotion and Stress-Tolerance-Enhancing Skills

Once the strains were isolated, we carried out characterization of them based on their abilities to beneficially interact with plants, as well as to potentially protect them against desiccation or drought events. Thus, we analyzed the plants’ survival rates under a full-desiccation process (xerotolerance), as well as their production of spores, biofilm, auxins, aminocyclopropane-1-carboxylic acid deaminase (ACCd) and siderophores, and their ability to solubilize phosphate. This characterization allowed us not only to select potentially beneficial strains, but also to analyze correlations with different environments in which they were originally isolated. As a control for the mentioned tests, we used *Eschericia coli* MC4100, and *Pseudomonas putida* KT2440 [65].

To characterize some of the plant-growth-promoting skills of the isolated strains, we employed a 96-well plate system. This system is based on a variation of conventional in vitro systems (glass tube), with a scaled down process volume, using flat-bottom microplates (Ratiolab, Dreieich, Germany). This allowed us to multiply the number of replicas in parallel, processing all samples at same time and reducing technical variations. Hence, the determination of biofilm, auxin and ACC deaminase production was carried out using this methodology. All these tests began by preparing an overnight culture of each strain in LB at 28 °C and 150 rpm. In order to clean nutrients from LB, these cultures were centrifuged at 14,000 rpm for 5 min, and resuspended in the same volume of sterile 0.45% NaCl solution in order to keep the OD value. Using this solution, we prepared each test by adjusting its optical density (OD) up to 0.05, in a volume of 200 µL per well. Each test was repeated 3 times, with 9 technical replicas (9 wells per strain). All these 96-well-plate tests were previously reported by other authors (properly cited below), with minor modifications.

#### 2.4.1. Drought Survival Mechanism: Xerotolerance vs. Sporulation

In order to evaluate the resilience and the main drought-survival mechanism, we carried out a sporulation vs. xeroprotectant production test, following Narvaez and collaborators’ (2010) indications, with minor modifications [66]. In brief, we prepared an overnight culture of each strain in LB medium, allowing them to grow up to 10^8^–10^9^ colony forming units (CFUs) per milliliter. A 300 µL aliquot of the culture was then centrifuged (14,000 rpm for 5 min) and resuspended in the same volume of sterile 0.45% NaCl. First, an aliquot of 100 µL was serially diluted (1:10) four times, and plated, drop-by-drop, in a sectioned LB plate. This aliquot was considered the control (mock). Then, to determine if the isolated strains were sporulants, another 100 µL aliquot was incubated at 72 °C for 30 min, and then, plated in drops in another section of the plate. Finally, the last 100 μL aliquot was placed in a sterile 24-well plate and dried under a current of sterile air for 72 h. Here, cells were resuspended in 100 µL sterile 0.45% NaCl solution, and serially diluted (1:10) four times before we placed the last dilution on the last section of the LB plate. The survival rate was calculated as CFUs/mL after drying, compared to the control. Any growth in the sporulation test section was considered a positive for sporulation. On the other hand, a strain showing a survival percentage >10%, but no growth after the sporulation test, was considered as a xerotolerant and, putatively, xeroprotectant producer.

#### 2.4.2. Biofilm Production

The biofilm production of the isolated strains was determined following the method described by Coffey and Anderson (2014) [67], with minor modifications. Briefly, a 0.05 OD_600nm_ culture of the strain was added to a well filled with 200 µL of LB. After incubation of the 96-well plate at 28 °C for 24 h and 150 rpm, planktonic bacteria were rinsed away with tap water, and the well-adhered structures were stained with 0.2% crystal violet for 20 min. Excess crystal violet was discarded and washed, as well, with tap water. To quantify the biofilm production, the stained structures were solubilized with 30% glacial acetic acid solution for 20 min, and measured at 550 nm.

#### 2.4.3. Auxin Production

To determinate if the isolated strains were able to produce auxins, the method described by Ambrosini and Passaglia (2017) [11] was followed, with slight modifications. Hence, a 0.05 OD_600nm_ culture of the strain was added to a well filled with 200 µL of an LB tryptophane-supplement medium (0.5 g/L of tryptophan). Then, the 96-well plate was incubated at 28 °C for 48 h and continuous shaking at 150 rpm. After incubation, the plates were centrifuged at 4000 rpm for 30 min, and 100 µL of the supernatant was transferred to a fresh plate. Finally, 100 µL of Salkowski reagent (0.5 M of FeCl_3_ and 35% HClO_4_) was added and the plate was incubated in darkness for 30 min at room temperature. Quantification was performed by measuring the solution at 530 nm, and the values were determined as indole-3-acetic acid (IAA) equivalents compared to a calibration curve.

#### 2.4.4. Aminocyclopropane-1-Carboxylic Acid (ACC) Deaminase Production

The production of aminocyclopropane-1-carboxylic acid (ACC) deaminase was determined through the growth of isolated strains in ACC medium. To prepare 100 mL of this medium, 10 mL of filter-sterilized 3 mM ACC, as the sole carbon and nitrogen source, was added to 90 mL sterile M9 salt solution (10× M9 stock: 33.7 mM of Na_2_·HPO_4_, 22 mM of KH_2_·PO_4_, 8.55 mM of NaCl; 1 mM of MgSO_4_; 0.3 mM of CaCl_2_; 1× trace elements: 0.031 mM of FeCl_3_·6 × H_2_O, 6.2 µM of ZnCl_2_, 0.76 µM of CuCl_2_·2 × H_2_O, 0.42 µM of CoCl_2_·2 × H_2_O, 1.62 µM of H_3_BO_3_, and 0.081 µM of MnCl_2_·4 × H_2_O). Then, 200 µL of this medium was pipetted in each well of a 96-well plate, and a 0.05 OD_600nm_ culture of the strain was added. After incubation of the 96-well plate at 28 °C for 72 h and 150 rpm, the growth of the strains was measured at 600 nm. A growth above 0.1 OD_600nm_ was considered positive, with the understanding that the strain was able to use ACC as a sole carbon and nitrogen source thought the production of ACC deaminase. 

#### 2.4.5. Siderophore Production

All isolated strains were screened for siderophore production using Blue Agar CAS medium, according to the indications of Louden and collaborators (2011) [68]. In brief, 10 µL of an overnight incubated culture in LB (28 °C, 150 rpm), previously centrifuged (5 min; 14,000 rpm) and resuspended in sterile solution of 0.45% NaCl, was placed as a single drop on Blue Agar Chrome Azurol S (CAS) plates, and incubated at 28 °C for 7 days. The appearance of a yellowish halo around the colonies indicated positive production of siderophores by the strains, and the diameter of the halo was measured from the center of the drop.

#### 2.4.6. Phosphate Solubilization

The phosphate solubilization skill of the isolated strains was tested following the indications of Nautiyal (1999) with slight modifications [69]. Thus, 10 µL of an overnight growth culture in LB (28 °C, 150 rpm), previously centrifuged (5 min; 14,000 rpm) and resuspended in sterile solution of 0.45% NaCl, was dropped in an NBRIP (per liter: glucose, 10 g; Ca_3_(PO_4_)_2_, 5 g; MgCl_2_·6H_2_O, 5 g; MgSO_4_·7H_2_O, 0.25 g; KCl, 0.2 g and (NH_4_)_2_SO_4_, 0.1 g). After seven days of incubation at 28 °C, the development of halos around the drops indicated positive phosphate solubilization. The diameter of the halos was measured, as well.

### 2.5. Plant Inoculation: Growth Promotion and Drought Tolerance Enhancement

For this test, we used lentil (*Lens culinaris* Medik.) ecotype ‘Beleza’ plants, kindly provided by the Instituto Nacional de Investigação Agrária e Veterinária (INIAV) in Elvas (Portugal). Fresh seeds were surface-sterilized using 70% ethanol for 3 min, and then, washed thrice with sterile double-distilled water. Thereafter, the seeds were incubated in water overnight at room temperature to improve the hydration and germination rate. After incubation, the seeds were placed on a wet paper towel inside a Magenta box. The paper was regularly moistened and replaced if an odor or microbial growth appeared. These boxes were placed in darkness at 22 °C for 2–3 days until most of the seeds had germinated, at which point they were transferred to 0.5 L pots full of a turf:vermiculite (3:1, *v/v*) mix. After a 24 h process of soil and greenhouse acclimatation, the seedlings were inoculated with 40 mL/pot of each candidate strain (after screening), consisting of up to 10^8^–10^9^ CFUs/mL (OD_600nm_ ≈ 1.0), in a 0.45% sterile NaCl solution as a carrier. Control plants (mock) were inoculated with the same volume of sterile 0.45% NaCl saline solution. Then, a set of plants (5 plants per treatment) was regularly irrigated (above 80% soil relative humidity (% SRH)); meanwhile, another was kept under watering restrictions (below 20% SRH). Ten days after treatment (DAT), the phenotype of the seedlings was recorded by measuring the root length, shoot height, root dry weight (DW) and total dry weight. Moreover, the number of secondary roots and average root thickness were measured using EZ-Root-VIS software (v2.5.4.0; University of Glasgow, Glasgow, UK) [70].

### 2.6. Statistics

Statistical analysis was performed in Prism (v9.0.0; GraphPad Software). Therefore, we applied the Student’s *t*-test and two-way ANOVA (with Tukey’s post-test) for pairwise and multi-group comparisons, respectively. The significance level was set at *p* < 0.05. The graphs were created in Prism and Excel 2019.

## 3. Results

### 3.1. Soil Characterization: pH, Electroconductivity and Water Content

The seed samples collected from the population of plants identified in the different locations were subjected to certain environmental growth conditions. Among them, soil and climate are those that most influence seeds characteristics, the number produced and their germination rate. Moreover, the seedborne microbiota may be significantly influenced, as well. In this way, since the soil is the main source of microorganism interactions with plants, its characteristics will allow us to better contextualize our results (Table 2). Hence, we first evaluated the climate conditions of the sampling locations. Taking into account historical precipitation, and that registered during 2022 (until the sampling time), we can observe clear differences between the control sampling location of ‘Oeiras’ and the others (almost double in most cases), with ‘Huércal-Overa’ as the most extreme case (10 times less precipitation). The following conditions affected the water content of the sampling points: Soil in ‘Oeiras’ recorded the highest humidity content (12.92%), while the rest of the samples were below 1.7%. The lowest values (0.24–0.25%) were recorded in ‘Carril-Busot’, ‘Baza’ and ‘Velez-Rubio’. Regarding the temperature, ‘Oeiras’ registered between 2–9 °C lower than the rest of the locations, whereas ‘Castejón de Monegros’ and ’Belchite’ showed the highest average temperature (approx. 35 °C).

Concerning the geological context, most of the locations were characterized by three main types of material. The most common geological contexts were sandstones/conglomerates (Castejón de Monegros, Belchite, Orce, Guadix, Lorca, Medinaceli, Bárdenas Reales and Agost) and marls/limestones (Oeiras, Vera, Baza, Carril-Busot, Huércal-Overa and Níjar), with a total of eight and six locations, respectively. Only the location ‘Vélez-Rubio’ was placed in the context of silicates (quartzites, phyllites and schists), but it also had residuary limestone. These contexts have given rise to soil types with a predominantly basic pH. Hence, we find that the most common types of soil along our sampling points (based on FAO classification) were Calcisols, with seven locations (Oeiras, Vera, Vélez-Rubio, Orce, Medinaceli, Níjar and Belchite). These soils are linked to arid and semi-arid regions due to the factors that originate from the accumulation of lime. Moreover, four other places were located in calcaric Fluvisols (Baza, Guadix, Lorca and Bárdenas Reales), located close to ancient riverbeds around Calcisols. Another type of soil linked to arid regions is Gypsisol, which was located at the ‘Castejón de Monegros’, ‘Carril-Bustot’ and ‘Agost’ sampling points. Here, instead of inorganic calcium compounds, the main precursor is gypsum. Finally, Cambisol was the main soil at the ‘Hércal-Overa’ sampling point, characterized by the accumulation of clay and iron oxides. This context explains how, in general, all sampling locations were determined to be alkaline, with a pH above 8, with only the control location, ‘Oeiras’, having a pH below 8.7. In contrast, the sampling location ‘Níjar’ was recorded as the most alkaline, with a pH of 9.69.

Finally, we evaluated the electroconductivity (EC) of these soil samples as an indirect salinity measure. Here, the ‘Oeiras’ sampling point showed the lowest EC value (0.15 dS/m). The rest of the locations, except of ‘Guadix’ (0.67 dS/m), showed EC values above 1. Among them, almost half (six) were characterized as saline soil (above 4 dS/m). Finally, the highest ECs were recorded in the ‘Agost’ and ‘Bárdenas Reales’ samples, with values above 5 (5.54 and 6.43 dS/m, respectively).

### 3.2. Germination Rate

The germination ratio of the seeds collected at the different sampling points was variable among the plant species and locations (Figure 2). Beginning with *M. sativa* seeds, the samples from ‘Oeiras’ were the ones that showed the lowest germination ratio (9.49%). Considering the rest of the sampling locations, the highest germination ratio was recorded in ‘Agost’ (49.8%), while ‘Belchite’ (13.36%) showed the lowest. In the case of the *B. bituminosa* seeds, Vera (5.68%) showed the lowest germination ratio, at almost half that of the recorded in ‘Oeiras’ (12.46%). Samples from ‘Huércal-Overa’, ‘Níjar’, ‘Lorca’ and ‘Vera’ showed a lower germination ratio than control location ‘Oeiras’. On the other hand, the highest germination ratios were shown by seeds collected in ‘Guadix’ and ‘Baza’ (34.93% and 32.95%, respectively).

### 3.3. Characterization of the Strains: Identification, Population Distribution and Comparative Analysis

In order to characterize the microbiota population, we first determined the number of colony-forming units (CFUs) per milligram of dry-weight (DW) biomass sampled (Figure 3). Considering the full population counted, the values for all the locations were higher than that of the control location (‘Oeiras’, 0.035 CFUs(×10^4^)/mg of DW biomass) in the case of *M. sativa* samples. The sample from ‘Orce’ was the one with smallest population recorded (0.065), but still double size of that in the control location; on the other hand, the location with largest population recorded was ‘Bárdenas Reales’ (1.62). For *B. bituminosa* samples, only two locations (‘Guadix’ and ’Vera’, 0.016 and 0.031) showed a smaller population than the control location (‘Oeiras’, 0.057 CFUs (×10^4^)/mg of DW biomass). Finally, the location with the largest population recorded was ‘Baza’ (2.05), 35 times larger than in control location, ‘Oeiras’.

In the collection of strains, we were able to isolate a total of 51 strains, from which up to 44 were finally identified (86.3%). We obtained a similar number of strains in *M. sativa* and *B. bituminosa* (25 and 26, respectively), but the seedborne microbiota in *M. sativa* seeds were almost 30% less diverse. Considering all strains identified except *Kosakonia cowanii* MS VR1-C and *Erwinia* sp. BB Ni-C, which may putatively be plant pathogens (less than 5% of the total strains identified), all the strains have been defined as plant-growth-promoting bacteria in the literature. Among them, about 60% were Gram- and 40% Gram+. Only 3 culturable strains were recovered from two kinds of seeds (*Enterobacter hormaechei*, *Pantoea agglomerans* and *Stenotrophomonas maltophilia*), which are both Gram-. Strains identified as *Paenibacillus polymyxa* only occurred in *M. sativa* seed samples.

Populations of microbiota associated with seeds used to be very variable. Under this consideration, in our test, we used seeds from several plants of the same sample area to be able to identify patterns of the occurrence and prevalence of strains in the populations (Figure 4a,b). Our studies showed that the most common strain detected in both types of seeds was *S. maltophilia*, with strains in *M. sativa* (36% of total) and 10 strains in *B. bituminosa* (38% of total), respectively. The strains of *P. polymyxa* were the second most prevalent in *M. sativa* seeds (32%), and together with the *Paenibacillus xylanexedens* strain, were shown to be very close in prevalence (nine strains) to *S. maltophilia*. In *B. bituminosa* seeds, we found about 16% of prevalence of the other *Paenibacillus* strains (4). Similarly, Enterobacteriaceae family strains had a prevalence of about 16% in *B. bituminosa* and 12% in *M. sativa* (3 strains). On the other hand, the Bacillaceae family (8%; 2 strains) was only present in *B. bituminosa*. Finally, the unidentified strains represented about 9% of the seed microbiota population of *M. sativa* (2), and 19% of *B. bituminosa* (5). All sequences were submitted and are accessible to GenBank through the accession number OP957072-OP957111.

In the collection of 51 isolated strains, a total of 16 unique species were identified (*A. hermannii*, *P. agglomerans*, *E. hormaechei*, *E. ludwigii*, *Erwinia* sp., *K. cowanii*, *S. maltophilia*, *B. cereus*, *B. nealsonii*, *P. megaterium*, *P. amylolyticus*, *P. pabuli*, *P. peoriae*, *P. polymyxa*, *P. tundrae* and *P. xylanexedens*), and only 7 could not be identified. Among them, eight were present in *M. sativa* seeds (*A. hermannii*, *P. agglomerans*, *E. hormaechei*, *K. cowanii*, *S. maltophilia*, *B. cereus*, *P. polymyxa* and *P. xylanexedens*) and 11 in *B. bituminosa* seeds (*P. agglomerans*, *E. hormaechei*, *E. ludwigii*, *Erwinia* sp., *S. maltophilia*, *B. nealsonii*, *P. megaterium*, *P. amylolyticus*, *P. pabuli*, *P. peoriae* and *P. tundrae*), sharing three strains among both (*P. agglomerans*, *S. maltophilia* and *P. polymyxa*), as mentioned above. After comparing the strains via phylogenetical analysis, some of the uninoculated strains (for which not enough base pairs were amplified to ensure their identification) were placed close to some of the other strains identified in this study. Hence, the strain MS GX-A was related to *Bacillus cereus*; moreover, BB CB-A, MS A-B and BB GX-A were related to *S. maltophilia* (Figure 4c). Considering the geographical location, the north isolation zone (‘Castejón de Monegros’, ‘Belchite’, ‘Bárdenas Reales’ and ‘Medinaceli’) provided a total of 13 strains, from which we identified six different species (*A. hermannii*, *E. hormaechei*, *S. maltophilia*, *B. cereus*, *P. polymyxa* and *P. xylanexedens*). Regarding the Levante isolation zone (‘Lorca’, ‘Agost’ and ‘Carril-Busot’), eight strains were isolated, with a total of five species identified (*E. hormaechei*, *S. maltophilia*, *B. nealsonii*, *P. peoriae* and *P. polymyxa*); on the other hand, in the south isolation zone (‘Guadix’, ‘Baza’, ‘Orce’, ‘Níjar’, ‘Vera’, ‘Huércal-Overa’ and ‘Vélez-Rubio’), we found up to 27 strains, of which we were able to identify 10 (*E. hormaechei*, *E. ludwigii*, *P. agglomerans*, *Erwinia* sp., *K. cowanii*, *S. maltophilia*, *P. megaterium*, *P. amylolyticus*, *P. tundrae* and *P. polymyxa*). Finally, in the Portugal isolation zone (‘Oeiras’), we managed to isolate three strains, and all of them were identified (*S. maltophilia*, *P. pabuli* and *P. polymyxa*).

These strains were contextualized considering the environmental conditions where the model plants grew. In this way, most of the strains (43) were isolated from plant seeds in soils with a pH ˃ 9. On the other hand, a total of 31 strains were isolated from soil with EC below 4 dS/m, while the rest of the strains were isolated from environments with a high salinity (˃4 dS/m) index. Moreover, up to 35 strains were isolated from plants growing in a soil with less than 0.75% water content, but only six species (*Paenibacillus amylolyticus*, *Pantoea agglomerans*, *Enterobacter hormaechei*, *Stenotrophomonas maltophilia*, *Kosakonia cowanii* and *Bacillus nealsonii*) were represented in culturable populations isolated from plants growing in soils with <0.25% water content. Interestingly, *B. nealsonii* was the only strain exclusively linked to Gypsisol soils. On their behalf, *S. maltophilia* and *P. polymyxa* were isolated independently of the soil pH, EC or water content, and even of the type of soil. However, only *S. maltophilia* was present in samples from all the geological contexts. On the other hand, one of the putative pathogens, *K. cowanii*, was only present in quartzite geological contexts. Finally, the strains *Erwinia* sp., *E. ludwigii*, *B. nealsonii*, *P. amylolyticus*, *P. tundrae* and *P. pabuli* were only isolated from marl/limestone geological context, and *P. peoriae*, *P. xylanexedens*, *B. cereus*, *P. megaterium* and *A. hermannii* were in sandstone/conglomerate context (see Appendix A).

Considering the locations where the strains where isolated, we performed a comparative analysis by plant species (Figure 5). Hence, in the *M. sativa* samples, we found that six locations presented *S. maltophilia*, and six presented *P. polymyxa*. Both strains were isolated together in three locations, ‘Belchite’, ‘Castejón de Monegros’ and ‘Bárdenas Reales’, all of them in the north of Spain (Ebro Basin region). Moreover, a total of seven locations presented Paenibacillus strains. Furthermore, the most diverse locations, with four strains, were ‘Guadix’ (two of them identified as S. maltophilia) and ‘Medinaceli’ (*Bacillus cereus*, *Enterobacter hormaechei*, *Paenibacillus xylanexedens* and *Stenotrophomonas maltophilia*). Finally, from ‘Velez-Rubio’, ‘Castejón de Monegros’ and ‘Bárdenas Reales’, we isolated three strains in each, but at least one of them was from the same species.

On the other hand, in *B. bituminosa* samples, we isolated *S. maltophilia* in eight locations, finding three strains from this species in ‘Vélez-Rubio’. Moreover, we isolated *Paenibacillus* strains from four locations, and *Enterobacteriaceae* strains from three locations. In this group, we identified *Enterobacter hormaechei* in two locations (‘Baza’ and ‘Lorca’). Additionally, four regions presented *Paenibacillus* strains (‘Oeiras’, ‘Baza’, ‘Níjar’ and ‘Lorca’). As a final consideration, the most diverse location seems to be ‘Baza’, with four strains (*Enterobacter hormaechei*, *Paenibacillus amylolyticus*, *Pantoea agglomerans* and *Stenotrophomonas maltophilia*), followed by ‘Guadix’, ‘Carril-Busot’, ‘Lorca’, ‘Vera’ and ‘Níjar’, with all of these locations having three strains. In the location of ‘Vélez-Rubio’ we also isolated three strains, but all of them were identified as *S. maltophilia*.

### 3.4. Screening for Plant-Growth-Promoting and Drought-Tolerance-Enhancement Skills

In order to evaluate *M. sativa* and *B. bituminosa* seedborne microbiota as candidates to develop new bioinoculants for plant-growth promotion and/or stress-tolerance enhancement in lentil plants, characterization of the collection of 51 isolated strains was carried out. In this context, we decided to use, when possible, a hybrid screening pipeline, using high-throughput qualitative/semi-quantitative (selective media/halo measuring) techniques. The results were compared with those obtained for teh control strains *E. coli* MC4100 and *P. putida* KT2440 (Appendix A). In brief, as shown in Table 3, considering the 51 strains isolated, about 3/4 were able to sporulate, while only 15% were characterized as xerotolerant. Moreover, 80% of the collection were characterize as biofilm producers, and almost half were able to produce more than 15 µg/mL of indoleacetic acid-equivalent auxins and grow notably in ACC medium. Finally, more than 60% of the collection strains were able to produce siderophores, and slightly more than 40% were able to solubilize inorganic phosphate.

#### 3.4.1. Main Stress-Survival Mechanism

The survival mechanism in bacterial strains can be very relevant to their interaction with plants, conditioning their resilience in the rhizosphere, their colonization ratio or the duration of such interactions. Thus, we decided to evaluate the two main mechanisms of survival against stresses (specifically drought): the production of spores and the production of xeroprotective compounds. Thus, by applying a heat-shock treatment, we were able to observe how almost 75% of the isolated strains were able to sporulate, with half of the collection able to achieve it at a high level (~45%). These levels were determined in terms of recovered population by comparing the growth rate with the control drops, as show in Figure 6a. On the other hand, only 15.7% were able to survive a full-desiccation process, without showing survival capacity via sporulation.

#### 3.4.2. Biofilm, Auxins and ACC Deaminase Production

The evaluation of some key skills for both stress-tolerance- and plant-growth-inducing processes, as well as biofilm, auxin and ACC deaminase production, were carried out based on a high-throughput 96-well plate system. The high number of replicas (n = 9) running simultaneously in the plate reader (as OD_550nm_) gave us a highly trustable quantification method to compare the collection strains’ performance with that of the control strains, *Escherichia coli* MC4100 (Ec-control) and *Pseudomonas putida* KT2440 (Pp-control). Beginning with biofilm production, all the strains showed a level of quantifiable biofilm structures above that shown by the Ec-control (0.22). Thus, only four strains (1, 28, 50 and 51) produced less biofilm, and the other eight strains (8, 13 19, 38, 41, 43, 46 and 49) showed a similar production level (no statistical difference); meanwhile, a total of 39 strains were producing biofilm structures with a statistically significant higher value than those in the Pp-control (0.82) (Figure 7a). In this last group, the strains *Stenotrophomonas maltophilia* BB GX-A, *Stenotrophomonas maltophilia* MS GX-C and *Stenotrophomonas maltophilia* MS GX-D (6, 11, 12) showed a biofilm production of 1.15 (>45%).

In the auxin production test, we also found that all strains of the collection were able to produce auxins above 10 µg/mL of IAA-equivalents, which was considerably higher than the Ec-control (1.02 µg/mL) (Figure 7b). However, only half of them produced ≥15. Among them, the strains *Paenibacillus polymyxa* MS O1-A, *Stenotrophomonas maltophilia* MS VR1-A, *Paenibacillus pabuli* BB Oeiras A and *Stenotrophomonas maltophilia* BB Oeiras B (28, 38, 45 and 49, respectively) were able to produce more than 21 µg/mL, and only *Paenibacillus pabuli* BB Oeiras A (27.1 µg/mL) produced a statistically higher amount than Pp-control (25.1 µg/mL).

Finally, as an indirect way to evaluate the production of the enzyme ACC deaminase, the strains were screened by culturing them in a minimal medium with 1-aminocyclopropane-1-carboxylate as the sole carbon and nitrogen source. Under these conditions, all the strains were able to grow more than 0.14 OD_600nm_ (Figure 7c). Among them, slightly more than half were able to grow more than 0.3 OD_600nm_. The strains *Atlantibacter hermannii* MS BE1-A, *Paenibacillus polymyxa* MS O1-B, *Bacillus cereus* MS M1-A and *Paenibacillus xylanexedens* MS M1-C showed a similar growth value to that shown by the Pp-control (~0.4 OD_600nm_).

#### 3.4.3. Siderophore Production and Phosphate Solubilization

The ability to produce siderophores, as well as to solubilize inorganic phosphate, were evaluated in the isolated strains (as in Figure 6b,c, respectively). The quantification of the colored halos or solubilization halos (mm), as a measure of the intensity or production ratio of the different strains, was compared with those produced by the control strains, *Escherichia coli* MC4100 (Ec-control) and *Pseudomonas putida* KT2440 (Pp-control). Despite more than 60% of the strains being able to produce siderophores (32 out 51), only around 20% of them were able to produce a statistically significantly larger halo than the Pp-control (2.23 mm) (Figure 7d). Within this group, the halos registered in the strains *Paenibacillus polymyxa* MS BR1-A, *Pantoea agglomerans* BB B2-B and *Stenotrophomonas maltophilia* BB B2-D (25, 35 and 37, respectively) were shown to be the largest (>4.2 mm), indicating more intense siderophore activity (almost double that of the Pp-control). Moreover, another 13 strains showed similar halo lengths to the one registered in the Pp-control.

On the other hand, among the 43% of the strains that were able to solubilize inorganic phosphate (22 out 51), 12 strains were able to show statistically significatly larger solubilization halos than the Pp-control (3.97 mm) (Figure 7e). For this group, the solubilization halos registered in the strains *Stenotrophomonas maltophilia* BB GX-A, *Stenotrophomonas maltophilia* MS GX-D and *Bacillus cereus* MS M1-A (6, 12 and 32, respectively) were shown to be the largest (>7.1 mm), indicating more intense phosphate-solubilization activity (almost double that of Pp-control). In the case of *Stenotrophomonas maltophilia* BB GX-A (8.4 mm), the halo length was double that registered in the Pp-control. Another four strains showed no statistical difference compared to the values registered in the Pp-control. In addition, in nine strains, a halo-like area of <0.5 mm was recorded, but they were not considered phosphate solubilizers due to the unclear result and minimal capacity they may show.

### 3.5. Plant Inoculation: Plant-Growth Promotion and Drought-Tolerance Enhancement

After carrying out the characterization of the strains, we evaluated which of them could be used as candidates to prepare inoculation tests in plants. Thus, despite the fact that some strains stood, out especially in some of their characteristics, we wanted to select as candidate strains those that met at least three of the characteristics evaluated in the screening (Appendix A). In addition, we wanted to select strains isolated from both types of model plants, giving more relevance to those strains that were found transversally in both species. Thus, using the 11 strains that showed the best values for each biochemical test, we selected the strains *Paenibacillus amylolyticus* BB B2-A (*Pa*B2A; ACC deaminase, biofilm and auxins), *Paenibacillus xylanexedens* MS M1-C (*Px*M1C; ACC deaminase, phosphate solubilization and xerotolerance), *Paenibacillus pabuli* BB Oeiras A (*Pp*OA; biofilm, auxins and phosphate solubilization), *Stenotrophomonas maltophilia* MS M1-B (*Sm*M1B; biofilm, phosphate solubilization and xerotolerance) and *Enterobacter hormaechei* BB B2-C (*Eh*B2C; biofilm, phosphate solubilization, siderophores and xerotolerance).

These strains were prepared as inoculants to treat lentil plants. Half of them were exposed to drought (<20% SRH), and the other half were continuously irrigated. Twelve days after treatment, the phenotypes of the plants were evaluated (Figure 8a–d). Hence, we observed that, under regular irrigation, plants inoculated with *Sm*M1B and *Pa*B2A showed a bigger size; in the case of the plants under water restrictions, plants inoculated with *Px*M1c, *Pa*B2A and *Pp*OA showed more vigorous growth than those under mock conditions. These visual differences are reflected in several quantified parameters. In terms of shoot height, plants inoculated with *Sm*M1B and *Pa*B2A under continuous irrigation were around 70% higher than the control set of plants; under drought conditions, treatment with *Sm*M1B was the most remarkable one, with registered plant sizes double those of the control set (Figure 8e). When both conditions were compared, only the set of plants treated with *Pp*OA showed bigger shoots (+37%) under drought than under regular irrigation. This is also reflected in the total dry weight (TDW), where plants treated with *Eh*B2C, *Sm*M1B and *Pp*OA increased by more than 20% compared to the control (24, 54 and 70%, respectively); however, a reduction in TDW was detected in some treatments under irrigation (*Px*M1c and *Pa*B2A). On the other hand, all the treatments applied under drought conditions showed an increased TDW by a minimum of 7% (*Px*M1C), which rose to almost double (+88%) in the case of the set of plants treated with *Pp*OA (Figure 8f). Interestingly, only the set of plants treated with *Pa*B2A showed a higher TDW (+46%) under drought than under regular irrigation.

In general, the root phenotype exposed clear architecture changes under the inoculant treatments. Thus, upon analyzing root dry weight (RDW), except in the *Pa*B2A treatment (−21%), all treatments underwent increased RDW under continuous irrigation compared to the control set, where the treatments with *Pp*OA and *Sm*M1B (+61% and +73%, respectively) were especially noticeable. Under drought conditions, all treatments except *Px*M1C and *Eh*B2C (no significant difference) showed an increased RDW by at least +20%, again, with plants undergoing *Pp*OA treatment showing higher values (+80%) (Figure 8g). After comparing both conditions, we detected the same or even more RDW under drought (even in the control set). Hence, the sets treated with *Pp*OA and *Pa*B2A showed the most relevant increases (+42% and +96%, respectively). To complete this root analysis, we evaluated the root length, the number of secondary roots and the average thickness of the main roots (Figure 8h–j). Under continuous irrigation, none of the sets of plants treated showed significant differences in root length compared to those in the control set of plants, except that treated with *Px*M1C (−16%); similarly, under drought conditions, all sets of treated plants showed no significant difference compared to the control set of plants, except those treated with *Pa*B2A, where the root length values were almost double. When we compared both conditions, in general, the tendency was to record shorter roots under drought, with only *Pa*B2A showing longer roots than under continuous irrigation (+12%). Regarding the number of secondary roots, under continuous irrigation, only the set of plants treated with *Sm*M1B showed significantly more, with a value more than double (+142%) that of the control set. On the other hand, under drought conditions, again, those plants treated with *Sm*M1B showed a significantly high number of secondary roots (+74%), but those treated with *Pp*OA and *Px*M1C exhibited the highest number of secondary roots (+121% and +305%, respectively). Comparing both applied conditions, in general, all the treatments conducted led to fewer secondary roots, except the sets treated with *Pp*OA and *Px*M1C, for which we consistently recorded more secondary roots (+50% and +108%, respectively). Finally, regarding the average root thickness, in the set of plants with continuous irrigation, we noticed that the values of those treated with *Px*M1C and *Eh*B2C were significantly lower than in the control set, while the set treated with *Pp*OA was the only one with significantly thicker roots. In the case of the set exposed to drought, only the plants treated with *Pa*B2A showed no significant difference. Considering the rest of the treatments, all the sets showed roots more than 25% thicker than the control set, with a relevant example being the set treated with *Px*M1C (+60%). When we compared both conditions, the roots were slightly less thick under drought conditions; however, in the sets treated with *Px*M1c and *Eh*B2C, we recorded thicker roots under this condition (+62% and +99%, respectively).

## 4. Discussion

Strains inherited through seeds represent a field of knowledge to be explored with enormous potential. Metagenomic analysis is increasingly able to distinguish clear patterns for these types of samples, which used to be highly variable. This offers us a new opportunity to find populations of interest in certain environments. In this way, populations subject to stress have been shown to be capable of transmitting populations of bacteria to improve the response and tolerance of the following generations [44]. This strategy may explain how natural populations adapt to variable environmental pressure conditions. Moreover, these mechanisms may also explain how a certain part of the population survives more often than others when faced with certain stresses, guaranteeing the species’ persistence [43,44,71]. Likewise, the high variability of bacterial populations that are usually found would thus respond to the proportion of the population that is capable of transmitting certain beneficial populations [43]. On the other hand, the impoverishment of microbiota in agricultural soils, together with the limitations imposed by plant domestication processes, condition the interactions that commercial plants, such as legumes, can handle [22,25,27]. This situation would explain the adaptability of the following generations through the transmission of beneficial populations.

This is the approach we addressed in order to select our study areas and model plants. Since our objective was to detect possible new treatments to improve resistance to increasingly arid conditions (drought and temperature increase), we selected populations of *Medicago sativa* and *Bituminaria bituminosa* that have adapted to arid zones [49,50,52,53]. The evaluation of the cultivable microbial populations of seeds produced under these conditions showed a series of patterns that fit with the hypothesis of population adaptability through the selection of inherited microbiota. In general, most of the isolated strains have been reported in the literature as beneficial, but some have been identified as pathogenic or possible plant pathogens. Starting with this group, considering the entire collection, very few potentially pathogenic plant strains (*Kosakonia cowanii* MS VR1-C (formerly *Enterobacter cowanii*) and *Erwinia* sp. BB Ni-C) were isolated [72,73]. In addition, their persistence and relative abundance in the different samples were very low compared to the rest of the strains. Their population numbers were about 0.011–0.017 CFU × 10^4^/mg of biomass DW, supposedly less than 2% of the sample’s total population. These types of strains are the ones that opened the door to the study of microbial populations in seeds, in order to control the spread of diseases through the use of foreign varieties. However, nowadays, there are increasing reports of the isolation of beneficial bacteria from seeds, as reported in the present work [74,75]. Moreover, the presence of both types of microorganism indicates that there are mechanisms to overcome the barriers that plants may face in the transfer to seeds. Thus, the persistence and relative abundance of these pathogens in seeds may be considered an indicator of the degree of health of the previous generation.

However, the largest populations and the most persistent in the different locations of this work showed a very different sign. The multiple samples showed a series of highly relevant beneficial populations, with the rest of the isolated strains previously described as promoters of plant growth. Thus, the strains *Bacillus cereus*, *Bacillus nealsonii*, *Priestia megaterium*, *Pantoea agglomerans*, *Enterobacter ludwigii*, *Stenotrophomonas maltophilia*, *Paenibacillus pabuli*, *Paenibacillus peoriae* and *Paenibacillus polymyxa* are highly recognized as growth promoters in different plant species [76,77,78,79,80,81,82,83,84,85,86,87,88]. Others, such as *Enterobacter hormaechei*, *Atlantibacter hermannii*, *Paenibacillus amylolyticus*, *Paenibacillus tundrae* and *Paenibacillus xylanexedens*, have less support in the literature, but have also been reported as beneficial or characterized as potentially beneficial due to their abilities, such as phytohormone production or the solubilization/mobilization of inaccessible nutrients [89,90,91,92,93,94]. On the other hand, most of these strains have been defined as enhancers of plant response and tolerance to stresses such as drought (*B. cereus*, *P. megaterium*, *E. ludwigii*, *P. agglomerans, P. amylolyticus* and *S. maltophilia*), salinity (*B. nealsonii*, *P. agglomerans*, *E. ludwigii* and *S. maltophilia*), high/low temperature (*B. cereus*, *P. tundrae* and *P. xylanexedens*), or heavy metals (*B. nealsonii*). Furthermore, some of them have been described as nematicides (*P. polymyxa*) and biocontrollers (*S. maltophilia*, *P. peoriae*, *P. polymyxa* or *P. xylanexedens*), assuming an additional adaptive defense against pathogens transferred to the next generation. Interestingly, most of the strains reported here (specially *Bacillus* sp., *Pantoea* sp. and *Paenibacillus* sp.) were also described previously as pathogen controllers inherited from seeds [74,76,95,96,97,98]. Additionally, some of these strains were previously identified as part of the seed microbiota in previous studies (*P. tundrae*, *P. megaterium* and *P. megaterium*). Likewise, other strains belonging to the genera *Paenibacillus* and *Enterobacter* were found in different populations associated with seeds [99,100,101].

After analyzing the geographical and soil contexts, no remarkable association was detected. The only exception could be found in the strains from ‘Oeiras’, as the higher-auxin-producing strains were those isolated from this place, identified as the wettest area. Despite auxins being very important in drought-tolerance in plant interaction processes, it is a very common phytohormone among plant-growth-promoting bacteria. This was the only skill in strains isolated from ‘Oeiras’ that showed up as the most relevant. In contrast, populations recorded in ‘Oeiras’ suggest that in places with fewer climatic restrictions, seeds may have inherited a lower level of the microbiota population, both in *M. sativa* and *B. bituminosa*. Only in some of the most arid regions were seeds shown to host fewer populations than the control location for the model plant *B. bituminosa*. Regarding the other abilities, it seems evident that biofilm production is a common feature in strains isolated from seeds in arid regions. This ability, in addition to favoring colonization and interaction mechanisms, seems to be related to root protection against soil stress [102,103]. Although this has been well described in conditions of pollution and salinity or against soil pathogens, its function as a protective cover under drought seems to have a similar meaning, adding moisture retention in cases of osmotic pressure or toxicity caused by oxidizing agents due to drought [104]. Some authors have described the protective capacity of biofilms in microorganisms under extreme environments, but few reports have directly linked the induction of tolerance to drought in plants to the production of biofilms [105,106]. Among them, most used to be linked to *Bacillus* strains, as reported in Wang et al., 2019, where they described this characteristic as fundamental in *Bacillus amyloliquefaciens* to induce tomato tolerance to drought [107].

The prevalence of siderophore producers and phosphate solubilizers is very high in the isolated strains. In the case of phosphate solubility, this ability has been described as fundamental in drought-tolerant bacteria that are beneficial to plants, since such environments usually have very low availability [108,109]. Similarly, the production of siderophores under stress conditions has been related to a necessary mechanism in degraded or depleted land. This is justified as a mechanism that facilitates the intake of iron in deficient terrain. Some authors even defend that the use of siderophore-producing rhizobacteria can improve plant growth in such environments [110]. However, some authors have described, for *Pseudomonas* species, that as soil moisture decreased, phosphate solubilizers increased, but siderophore producers decreased [111]. Thus, it seems that phosphate solubilization is more relevant, but the seedborne strains we isolated appear to be able to assemble even more desirable capacities to improve plant tolerance in arid environments.

On the other hand, the production of ACC deaminase is one of the most common characteristics among bacteria that improve plant tolerance to stressful environments. In addition, it is important to note that it is one of the best conserved characteristics within seed microbiota [71]. Similarly, sporulation is another of the most conserved traits in seed microbiota. The need to persist in the dehydrated environment of the mature seed seems to have made this ability necessary [112]. Only 15% of the strains isolated in this work used only xerotolerance as a method of survival in drought, so sporulation may be a more common ability in heritable microbiota in arid regions, as we observed in our strains. Despite this, xerotolerant capacities are especially interesting for the survival of the plant, and are an extra capacity to consider.

Regarding the possibility of using some of these strains as inoculants to promote growth and improve the response to drought in commercial crops, the strains we selected to treat lentils obtained a range of very interesting results. These revealed that the treatments shared some phenotypes, but each one caused evidently different responses. The tests were finished when the first wilting symptoms were evident in order to better evaluate types of plants whose tissues are easily damaged by stress. Thus, the plants treated with the strains *Sm*M1B and *Pa*B2A were shown to be bigger under continuous irrigation, and the set treated with *Pa*B2A was also shown to be bigger under drought. Specifically, the treatment with the *Sm*M1B strain stood out in terms of stem height reached, while the plants treated with the *Pa*B2S strain stood out in terms of total dry weight under continuous irrigation conditions. On the other hand, in general, the weight of the root increased with bacterial treatment under continuous irrigation and drought, but the size was lower under continuous irrigation. In this sense, the plants treated with *Px*M1C and *Pp*OA formed a more complex architecture than the those undergoing the other treatments. Conversely, the treatment with the strains *Sm*M1B, *Pa*B2Ae and *Eh*B2C showed bipartite and simpler root systems. In general, plants treated with bacteria tend to show more developed root systems. However, in drought conditions, this situation is highly variable, and it may be more effective to modulate the architecture towards smaller but more numerous roots [113,114,115]. This situation is highly determined by the type of soil and environment in which the plant grows.

An interesting case is the set of plants treated with *Paenibacillus peoriae* OA, which stands out in many parameters evaluated in drought. This strain comes from a less restrictive environment, and thus, may seem less likely to have been selected for improved drought tolerance. It is also necessary to consider, here, that auxins are a described mechanism both for promoting growth and for improving drought tolerance in plants [116]. In addition, the portion of strains that we selected in drought zones showed great performance under these conditions, despite not being especially outstanding in the production of auxins. This may indicate that the seeds may harbor enough diversity of microorganisms to give sufficient alternatives to the next generation, despite the environmental conditions they have to face.

Considering those strains selected from more arid environments, despite the fact that *Paenibacillus amylolyticus* B2A is not often referred to in the literature, here, we can see how this strain may be a good treatment to promote growth and improve tolerance to drought in plants. In the case of *Stenotrophomonas maltophilia* M1B, there are many reports of its capacity to induce higher shoots and longer roots, even under salt conditions; however, here, we show one of the few reports that this strain induces drought tolerance [117], and the first evaluation of their effect as root architecture enhancers, especially under drought conditions. Finally, plants treated with the strains *Paenibacillus xylanexedens* M1C and *Enterobacter hormaechei* B2C showed an interesting root thickness pattern, with thinner roots during continuous irrigation, and thicker roots than with any other treatment during drought. This could indicate the relevance of bacteria-mediated root morphology adaptation to different environmental conditions [113].

These results anticipate that, by studying and testing strains isolated from seeds collected in arid environments, we may provide new candidates for drought-tolerance treatments in closely related plants. This is a very interesting opportunity to solve new production challenges and fight against fatigue caused by different stresses in modern crops. Hence, the use of seed-associated microbiota could result in a great step forward in the use of beneficial microorganisms as plants inoculants. In this regard, new trials should be carried out to better define their populations, and even provide better knowledge of the mechanisms involved.

## 5. Conclusions

In this work we evaluated the use of culturable seedborne microbiota from wild, drought-adapted legumes as a new source of plant drought-tolerance enhancers. We found that main populations in culturable seedborne microbiota were previously described as plant-beneficial bacteria, with a minimal presence of putative pathogens. This proportion revealed a clear prevalence of beneficial interacting bacteria in the inherited populations. Further assays should be performed to understand the filtering mechanisms involved. Among the isolated strains, the production of biofilm, auxins and ACC deaminase were remarkably common skills; moreover, the vast majority were able to solubilize phosphates and produce siderophores. These mechanisms can be considered some of the preferential skills for inheritable microbiota. Furthermore, the predominance of sporulation over other xerotolerant mechanisms revealed this mechanism to be the most common among seed microbiota for surviving total desiccation.

On the other hand, different strains of *Stenotrophomonas maltophilia* and *Paenibacillus polymyxa* were isolated in both model plants despite their diverse origins and environmental contexts, suggesting a conserved or common inherited microbiota. Moreover, *Enterobacter hormaechei* was also present in seeds of both model plants, but they were restricted to some sampling contexts. Considering the strains tested as inoculants, we conclude that the strains *Paenibacillus amylolyticus* B2A and *Stenotrophomonas maltophilia* M1B stand out in lentil treatments. Interestingly, most of the strains tested were able to cause relevant changes by inducing shorter root length and a more complex root architecture, which were more evident under drought treatments. Thus, the use of bacteria isolated from the seeds of wild legumes adapted to arid contexts was successfully tested to induce drought tolerance and growth promotion in commercial lentils. New assays are required that include treatments with consortia, which may offer more solid results and better resistance of plants to drought events.

## Figures and Tables

**Figure 1 biology-11-01838-f001:**
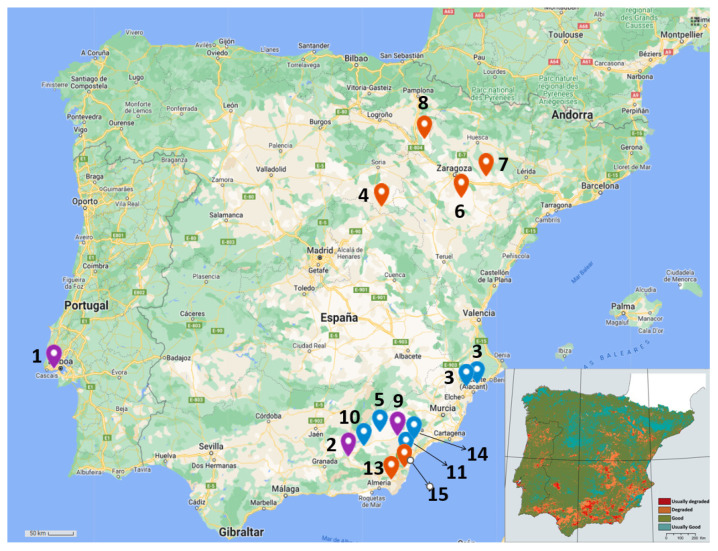
**Sampling locations.** The upper map (prepared in My Maps, Google) shows the location selected for sampling. Numbers refer to Table 1 sample denomination. Location pins in red indicate samples of alfalfa (*Medicago sativa*); blue pins, samples of pitch clover (*Bituminaria bituminosa*); and violet pins, samples of both types of model plants. Lower-right map indicates risk of degradation by desertification (modified from del Barrio et al., 2010) [59]. Here, red-colored areas indicate land that is usually degraded; orange, land that is degraded; green, land that is in good condition; and blue, land that usually has no risk or is usually in good condition.

**Figure 2 biology-11-01838-f002:**
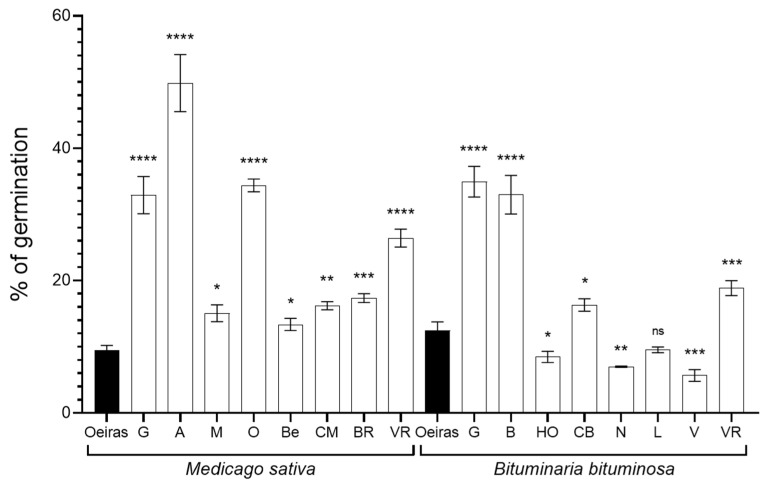
**Germination ratios.** The bar graph shows the germination rate in % from three independent experiments (n = 30) recorded in alfalfa (*Medicago sativa*) and pitch clover (*Bituminaria bituminosa*), collected in the natural locations. The ‘Oeiras’ location (in black), for both species, was considered a control representing a humid/semi-humid location in contrast with the rest of locations. The locations are abbreviated as follows: ‘G’, Guadix; ‘A’, Agost; ‘M’, Medinaceli; ‘O’, Orce; ‘Be, Belchite; ‘CM’, Castejón de Monegros; ‘BR’, Bárdenas Reales; ‘VR’, Vélez-Rubio; ‘B’, Baza; ‘HO’, Huércal-Overa; ‘CB’, Carril-Busot; ‘N’, Níjar; ‘L’, Lorca; and ‘V’, Vera. The sets of data were compared using a two-tailed Student’s *t*-test and 95% confidence intervals, where the asterisks represent a statistically significant difference at *p* < 0.05, *; *p* < 0.01, **; *p* < 0.001, ***; and *p* < 0.0001, ****, compared to their respective control (‘Oeiras’ location); ^ns^ stands for statistically non-significant. Error bars represent s.d.

**Figure 3 biology-11-01838-f003:**
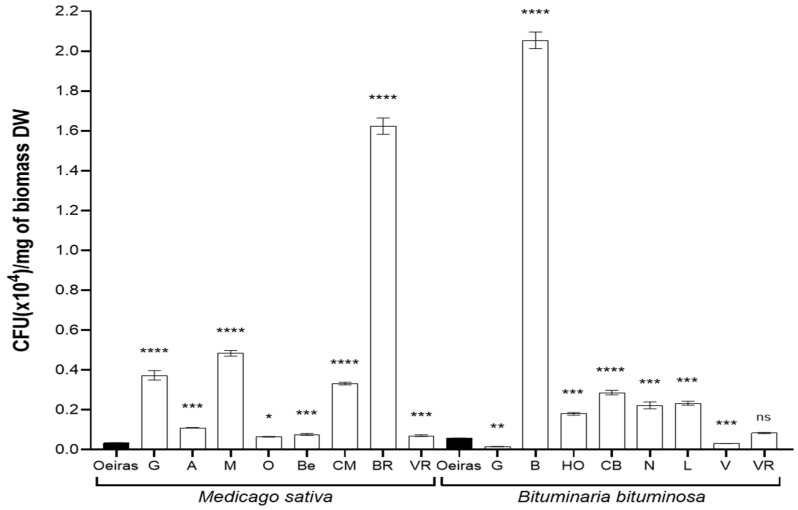
**Total population of culturable microbiota isolated from *Medicago sativa* and *Bituminaria bituminosa* seeds.** The bar graph indicates the culturable population recorded in the different locations from three independent experiments (n = 6) recorded in alfalfa (*Medicago sativa*) and pitch clover (*Bituminaria bituminosa*), collected in the natural locations. The ‘Oeiras’ location (in black), for both species, was considered a control representing a humid/semi-humid location in contrast with the rest of locations. The locations are abbreviated as follows: ‘G’, Guadix; ‘A’, Agost; ‘M’, Medinaceli; ‘O’, Orce; ‘Be, Belchite; ‘CM’, Castejón de Monegros; ‘BR’, Bárdenas Reales; ‘VR’, Vélez-Rubio; ‘B’, Baza; ‘HO’, Huércal-Overa; ‘CB’, Carril-Busot; ‘N’, Níjar; ‘L’, Lorca; and ‘V’, Vera. The sets of data were compared using a two-tailed Student’s *t*-test and 95% confidence intervals, where the asterisks represent a statistically significant difference at *p* < 0.05, *; *p* < 0.01, **; *p* < 0.001 ***; and *p* < 0.0001, ****; ^ns^ stands for statistically non-significant. Error bars represent s.d.

**Figure 4 biology-11-01838-f004:**
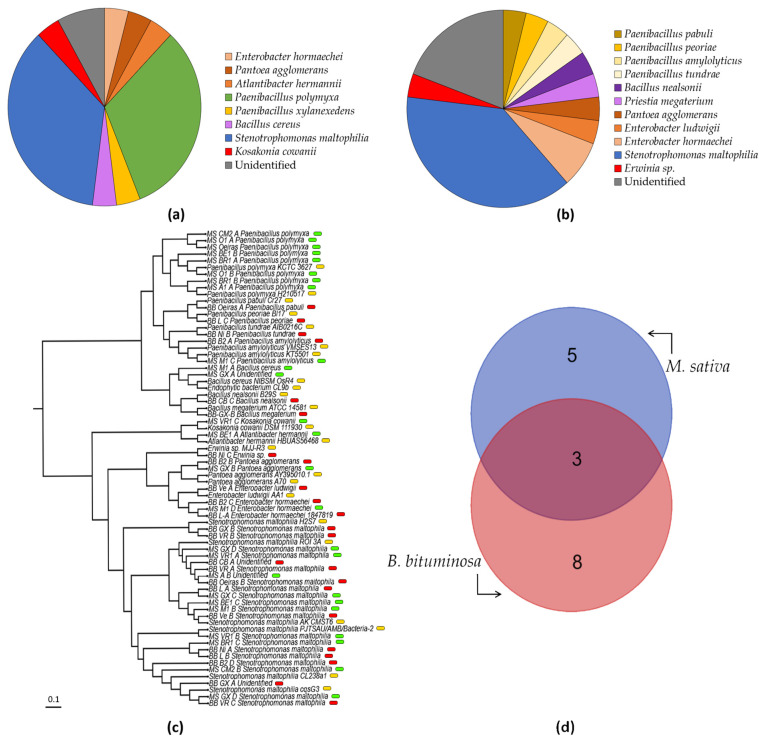
**Prevalence of strains isolated from *Medicago sativa* and *Bituminaria bituminosa* seeds.** The pie charts show the proportion of each strain isolated by sample location in (**a**) *Medicago sativa* and (**b**) *Bituminaria bituminosa* seeds. Colors of each section stand for *Stenotrophomonas maltophilia* (blue), *Paenibacillus polymyxa* (green), other *Paenibacillus* strains (yellow), *Enterobacteriaceae* strains (orange), *Bacillaceae* strains (purple), *putative pathogens Erwinia* sp. and *Kosakonia cowanii* (red), and unidentified strains (grey). The phylogenetic tree (**c**) shows the related species used to identify our collection, where green labels stand for *M. sativa* strains; red, for *B. bituminosa*; and yellow, for the control strains determined by BLAST for identified strains. The Venn diagram (**d**) shows the number of species isolated from *M. sativa* and *B. bituminosa* seeds, and their overlapping.

**Figure 5 biology-11-01838-f005:**
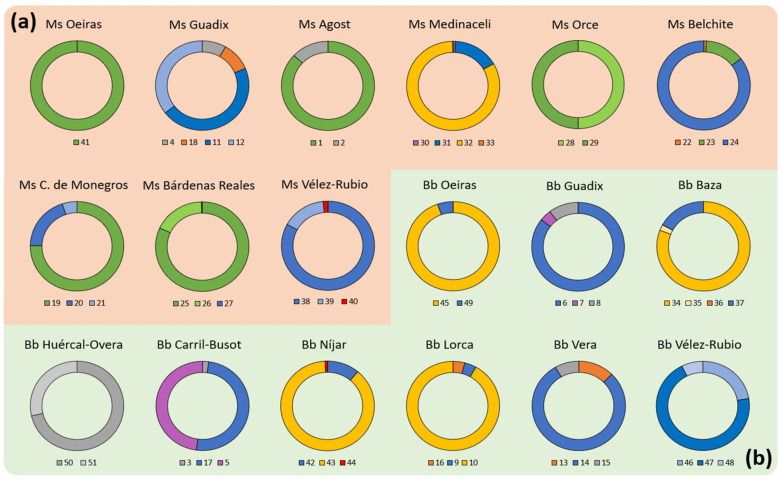
**Population distribution in communities isolated from *Medicago sativa* and *Bituminaria bituminosa* seeds.** The circle pie charts show the proportion of each strain isolated by sample location in (**a**) *Medicago sativa* (orange-background group) and (**b**) *Bituminaria bituminosa* (green-background group) seeds. Colors of each section represent Stenotrophomonas maltophilia (blue), *Paenibacillus polymyxa* (green), other *Paenibacillus* strains (yellow), *Enterobacteriaceae* strains (orange), *Bacillaceae* strains (purple), putative pathogens *Erwinia* sp. and *Kosakonia cowanii* (red), and unidentified strains (grey). If two or more strains identified in same sample belong to the same species, the range of intensity of the color (light to dark) is included, with a darker tone representing a larger proportion compared to the total population. Data shown is representative of three independent experiments (n = 6).

**Figure 6 biology-11-01838-f006:**
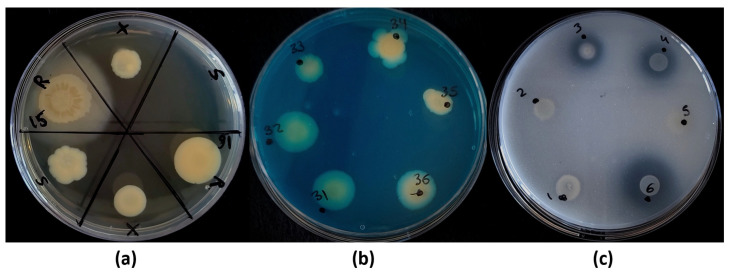
**Petri dish-based screening methods.** Plates prepared to discern (**a**) the main survival mechanism (LB plates); (**b**) siderophore production (Blue Agar CAS plates); and (**c**) solubilization of inorganic phosphate (NBRIP plates). In the first case, ‘R’ stands for regular or mock conditions (control); ‘X’, for the culture being resuspended in a full-desiccation process; and ‘S’, for the culture after thermal shock treatment. Strains were defined as sporulant if growth was detected in the ‘S’ section (as in strain 16 in the example plate); on the other hand, strains growing only in the ‘X’ section, but not in the ‘S’ section (as in strain 15 of the example plate), were considered to survive via the production of xeroprotective compounds (xerotolerant strains). Additionally, the growth ratios of the ‘X’ and ‘S’ treatments were compared to the control (‘R’) as an indirect way to evaluate the survival efficiency (see Table 3). In the second case, yellowish/orangish-stained colonies were considered positive in siderophore production (as in strains 34, 35 and 36 of the example plate); the staining halos around the colonies were measured (mm). In the last case, the solubilization halo around colonies was considered positive in phosphate solubilization (as in strains 3, 4 and 6 of the example plate).

**Figure 7 biology-11-01838-f007:**
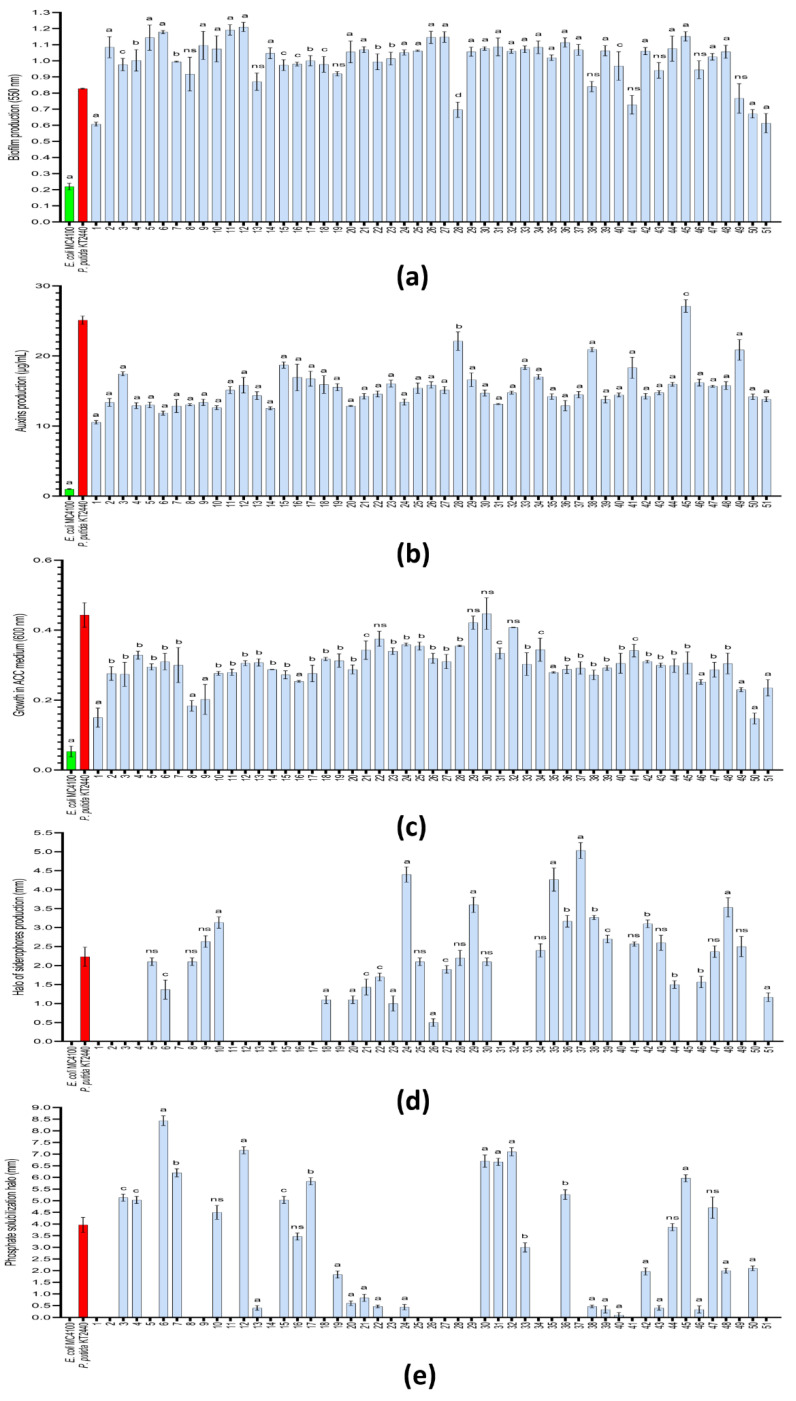
**Comparison of biochemical tests carried out on the isolated strains.** The bar graphs show the production of (**a**) biofilm and (**b**) auxins; (**c**) the growth recorded in ACC medium as the sole nitrogen and carbon source (indicative of ACC deaminase production); (**d**) the production of siderophores (mm of halo); and (**e**) the solubilization of phosphate (mm of halo), from three independent experiments (n = 3). The numbers of each column indicate the strain evaluated (see Table 3). The column in green represents *E. coli* (control), and the column in red represents *P. putida* (control). The sets of data were compared using a two-way ANOVA test and 95% confidence intervals (with Tukey’s post-tests), where letters indicate a statistically significant difference at ^a^ *p* < 0.0001; ^b^ *p* < 0.001; ^c^ *p* < 0.01 and ^d^ *p* < 0.01, compared to the values recorded in the main control strain, *P. putida* KT2440; ^ns^ stands for statistically non-significant. Error bars represent s.d.

**Figure 8 biology-11-01838-f008:**
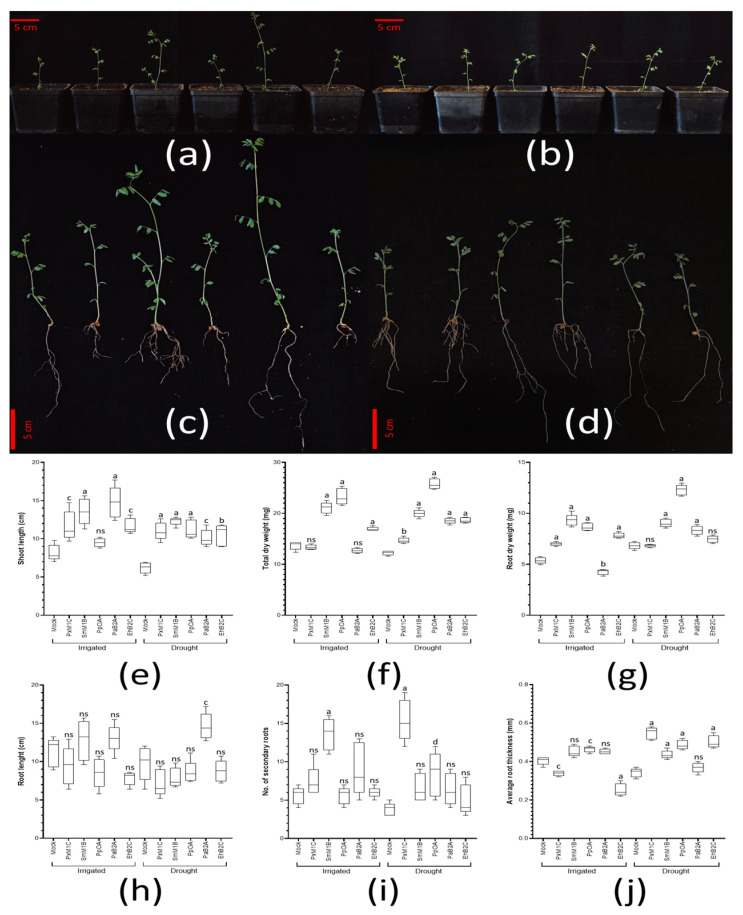
**Plant-growth promotion and drought-tolerance enhancement via inoculant treatment.** The pictures show the side view and the full plant view of treated seedlings under (**a**,**c**) continuous irrigation, and (**b**,**d**) water restriction (drought). The box plot graphs show the (**e**) shoot height; (**f**) root length; (**g**) number of secondary roots; and (**h**) average root thickness. Finally, the (**i**) root dry weight, and (**j**) total dry weight were recorded for each treatment and condition applied. Four independent experiments (each one with n = 5 biologically independent samples) were prepared in this test. Whiskers represent the minimum to maximum data range, and the median is represented by the central horizontal line. The upper and lower limits of the box outline represent the first and third quartiles. Here, ‘Mock’ stands for the control set of seedlings. ‘*Px*M1C’ represents seedlings treated with *Paenibacillus xylanexedens* MS M1-C; ‘*Sm*M1B’, with *Stenotrophomonas maltophilia* MS M1-B; ‘*Pp*OA’, with *Paenibacillus pabuli* BB Oeiras A; ‘*Pa*B2A’, with *Paenibacillus amylolyticus* BB B2-A; and ‘*Eh*B2C’, with *Enterobacter hormaechei* BB B2-C. The sets of data were compared using a two-way ANOVA test and 95% confidence intervals (with Tukey’s post-tests), where letters indicate a statistically significant difference at ^a^ *p* < 0.0001; ^b^ *p* < 0.001; ^c^ *p* < 0.01; and ^d^ *p* < 0.01; ^ns^ stands for statistically non-significant, compared to the mock set in each condition (irrigation and drought).

**Table 1 biology-11-01838-t001:** **Sampling locations.** Locations are here defined by their number (for map location), code, location, province and country (where PT means Portugal, and SP, Spain). GPS coordinates and altitude of the sampling area are approximative.

Alfalfa (*Medicago sativa*)					Pitch Clover (*Bituminaria bituminosa*)					
Code	Location	Province (Country)	GPS	Altitude (mamsl)	#	Code	Location	Province (Country)	GPS	Altitude (mamsl)
**Oeiras**	Oeiras	Lisbon (PT)	38°42′32.8″ N 9°16′02.3″ W	60	**1**	Oeiras	Oeiras	Lisbon (PT)	38°42′32.8″ N 9°16′02.3″ W	60
**G**	Guadix	Granada (SP)	37°19′03.1″ N 3°07′19.0″ W	957	**2**	G	Guadix	Granada (SP)	37°19′03.1″ N 3°07′19.0″ W	957
**A**	Agost	Alacant (SP)	38°26′15.6″ N 0°38′02.7″ W	294	**10**	B	Baza	Granada (SP)	37°30′11.2″ N 2°47′08.1″ W	890
**M**	Medinaceli	Soria (SP)	41°10′11.4″ N 2°25′16.9″ W	1022	**11**	HO	Huércal-Overa	Almería (SP)	37°24′12.0″ N 1°56′16.4″ W	278
**O**	Orce	Granada (SP)	37°44′01.8″ N 2°30′50.6″ W	902	**12**	CB	Carril-Busot	Alacant (SP)	38°28′38.2″ N 0°25′18.7″ W	295
**Be**	Belchite	Zaragoza (SP)	41°17′55.4″ N 0°44′37.2″ W	440	**13**	N	Níjar	Almería (SP)	36°57′19.3″ N 2°12′21.8″ W	282
**CM**	Castejón de Monegros	Huesca (SP)	41°37′12.0″ N 0°14′04.7″ W	479	**14**	L	Lorca	Murcia (SP)	37°38′05.9″ N 1°44′24.9″ W	356
**BR**	Bárdenas Reales	Navarra (SP)	42°10′34.0″ N 1°31′01.8″ W	294	**15**	V	Vera	Almería (SP)	37°10′13.5″ N 1°56′43.5″ W	127
**VR**	Vélez-Rubio	Almería (SP)	37°38′50.5″ N 2°05′12.3″ W	845	**9**	VR	Vélez-Rubio	Almería (SP)	37°38′50.5″ N 2°05′12.3″ W	845

mamsl stands for meters above mean sea level.

**Table 2 biology-11-01838-t002:** **Climatic, geologic and edaphic conditions at sampling locations.** This table compiles values of climatic conditions (precipitation and temperature), geologic context and soil variables (type, pH, electro-conductivity and water content), which are factors that may condition the presence/absence and diversity of plant-interacting microbiota.

		Climatic and Soil Conditions of Sampling Location								
#	Location Code ^a^	Average Precipitation (mm) ^b^	Precipitation till July, 2022 (mm) ^c^	Average Temperature in Summer (°C) ^d^	Average Teperature in Summer 2022 (°C) ^d^	General Geological Context ^e^	Soil Type (FAO Classification) ^f^	pH ^g^	EC ^g,h^ (dS/m)	Water Content (%) ^g^
**1**	Oeiras	591	301	20.2 [23.4]	24.2 [26.4]	Limestones	Calcic luvisol/calcisol	8.04	0.15	12.92
**2**	G	290	184	25.5 [30.4]	23.8 [28.5]	Sandstones/ conglomerates	Calcaric fluvisol	9.20	0.67	0.73
**3**	A	345	245	24 [28.5]	24.2 [32.3]	Conglomerates/ sandstones/ claystones/silstones	Haplic gypsisol	9.27	5.54	0.70
**4**	M	489	136	19.8 [26.4]	24.5 [30.9]	Conglomerates/ sandstones/silstones/ claystones	Cambic calcisol	9.30	1.02	1.68
**5**	O	366	144	22.0 [28.6]	22.3 [27.8]	Sandstones/ conglomerates	Cambic calcisol	9.38	3.05	0.30
**6**	Be	354	130	22.5 [28.5]	27.7 [34.9]	Sandstones/claystones/conglomerates	Haplic calsicol	9.06	4.21	1.45
**7**	CM	341	56	23.4 [30.0]	27.6 [35.1]	Conglomerates	Haplic gypsisol /calcisol	9.29	5.62	1.46
**8**	BR	302	211	21.6 [28.2]	26.3 [34.0]	Conglomerates/ sandstones/ silstones/claystones	Calcaric fluvisol/ Haplic calcisol	8.46	6.43	1.25
**9**	VR	238	58	21.5 [27.8]	26.5 [33.1]	Quartzite/ phyllites/ schists/limestones	Cambic calcisol	9.18	4.87	0.25
**10**	B	335	195	22.7 [29.9]	30.2 [31.3]	Marls/limestones	Calcaric fluvisol/ leptosol	9.38	1.34	0.25
**11**	HO	244	36	20.9 [27.5]	25.8 [33.5]	Marls/marly-limestones/ sandstones	Cambic cambisol	9.26	3.22	0.62
**12**	CB	421	267	24.2 [32.3]	24.2 [32.3]	Marls/limestone/ sandstones	Haplic gypsisol	9.30	4.08	0.24
**13**	N	223	77	24.0 [28.7]	27.2 [29.9]	Marls/ sandstones	Haplic calcisol	9.69	1.78	0.40
**14**	L	280	89	27.9 [35.6]	23.8 [29.4]	Conglomerates/ sandstones	Eutric cambisol/ Calcaric fluvisol	8.77	2.67	0.37
**15**	V	278	68	24.5 [28]	25.9 [29.3]	Marls	Cambic calcisol	9.39	1.24	0.50

^a^, average precipitation in last 20 years; ^b^, accumulated precipitation in 2022 (January to July); ^c^, average temperature in last 20 years between June and September, with average maximal temperature in brackets; ^d^, average temperature in 2022 between June and September, with average maximal temperature in brackets *(Note: these data were collected at the closest meteorological station provided by the Instituto Português do Mar e da Atmosfera (IPMA), and the Agencia Estatal de Meteorología (AEMET*–*Open Data), September-2022*); ^e^, data provided by the Mapa Geológico Nacional of Spain (MAGNA, 1:50.000) from the Instituto Geológico y Minero de España (IGME), and by the Portuguese Geologic maps 1:25.000 from the Laboratório Nacional de Energia e Geologia (LNEG); ^f^, data provided by the Cartografía dos Solos-Escala 1:25.000 (Norte) in the Sistema Nacional de Informação do Solo (SNIS) form the Direcção-Geral de Agricultura e Desenvolvimento Rural (DGADR), and by the Mapas Edafológicos and Atlas Nacional de Suelos (1:100,000) from the Instituto Geográfico Nacional (IGN); ^g^, values of samples collected in 10–15 cm deep fraction of soil; ^h^, ‘EC’ stands for Electric Conductivity measurement.

**Table 3 biology-11-01838-t003:** Strains isolated from *M. sativa* and *B. bituminosa* seeds. List of strains isolated and identified in this study by model plant and location. This table also includes a summary of the plant-growth-promoting skills of each isolate, where ‘Bfm.’ Stands for biofilm structure production; ‘Aux.’, auxin production; ‘ACCd’, growth in ACC medium to determine ACC deaminase production; ‘P. sol.’, phosphate solubilization; ‘Sdr.’, siderophore production; ‘Spr.’, sporulation; and ‘Xer.’, xerotolerance. For some tests, a range-mark was prepared: in biofilm (550 nm reading), ‘+’ stands for values < 0.7; ‘++’, for values between 0.71–1.14; and ‘+++’, for values ˃ 1.15. In auxins (µg/mL), ‘+’ stands for values < 14; ‘++’, for values between 15 and 20; and ‘+++’, for values ˃ 21. For growth in ACC medium (600 nm reading), ‘+’ stands for values < 0.2; ‘++’, for values between 0.21 and 0.4; and ‘+++’, for values ˃ 0.4.

#	Isolate Code	Scientific Name	Plant	Sampling Location	Skill Screening						
					Bfm.	Aux.	ACCd	P.sol.	Sdr.	Spr.	Xer.
**1**	MS A-A	*Paenibacillus polymyxa*	*M. sativa*	Agost	-	+	-	-	-	++	-
**2**	MS A-B	Unidentified	*M. sativa*	Agost	++	+	+	-	-	+++	+
**3**	BB CB-A	Unidentified	*B. bituminosa*	Carril-Busot	+	++	+	+	-	+++	-
**4**	MS GX-A	Endophytic bacterium	*M. sativa*	Guadix	++	+	++	+	-	+++	+
**5**	BB CB-C	*Bacillus nealsonii*	*B. bituminosa*	Carril-Busot	+++	+	+	-	+	+++	+
**6**	BB GX-A	*Stenotrophomonas maltophilia*	*B. bituminosa*	Guadix	+++	+	++	+	+	-	+
**7**	BB GX-B	*Priestia megaterium*	*B. bituminosa*	Guadix	+	+	++	+	-	+++	+
**8**	BB GX-C	Unidentified	*B. bituminosa*	Guadix	+	+	-	-	+	+	+
**9**	BB L-B	*Stenotrophomonas maltophilia*	*B. bituminosa*	Lorca	++	+	+	-	+	-	+
**10**	BB L-C	*Paenibacillus peoriae*	*B. bituminosa*	Lorca	++	+	+	+	+	++	+
**11**	MS GX-C	*Stenotrophomonas maltophilia*	*M. sativa*	Guadix	+++	++	+	-	-	-	+
**12**	MS GX-D	*Stenotrophomonas maltophilia*	*M. sativa*	Guadix	++	++	++	+	-	-	+
**13**	BB Ve-A	*Enterobacter ludwigii*	*B. bituminosa*	Vera	-	+	++	-	-	-	+
**14**	BB Ve-B	*Stenotrophomonas maltophilia*	*B. bituminosa*	Vera	++	+	+	-	-	-	+
**15**	BB Ve-C	Unidentified	*B. bituminosa*	Vera	+	++	+	+	-	-	+
**16**	BB L-A	*Enterobacter hormaechei*	*B. bituminosa*	Lorca	+	++	+	+	-	-	+
**17**	BB CB-B	*Stenotrophomonas maltophilia*	*B. bituminosa*	Carril-Busot	++	++	+	+	-	-	+
**18**	MS GX-B	*Pantoea agglomerans*	*M. sativa*	Guadix	+	++	++	-	+	-	+
**19**	MS CM2-A	*Paenibacillus polymyxa*	*M. sativa*	C. de Monegros	+	++	++	+	-	++	+
**20**	MS CM2-B	*Stenotrophomonas maltophilia*	*M. sativa*	C. de Monegros	++	+	+	-	+	-	+
**21**	MS CM2-C	*Stenotrophomonas maltophilia*	*M. sativa*	C. de Monegros	++	+	++	+	+	-	+
**22**	MS BE1-A	*Atlantibacter hermannii*	*M. sativa*	Belchite	+	+	++	-	+	-	+
**23**	MS BE1-B	*Paenibacillus polymyxa*	*M. sativa*	Belchite	++	++	++	-	+	++	+
**24**	MS BE1-C	*Stenotrophomonas maltophilia*	*M. sativa*	Belchite	++	+	++	-	+	-	+
**25**	MS BR1-A	*Paenibacillus polymyxa*	*M. sativa*	Bárdenas Reales	++	++	++	-	+	+++	+
**26**	MS BR1-B	*Paenibacillus polymyxa*	*M. sativa*	Bárdenas Reales	+++	++	++	-	-	++	+
**27**	MS BR1-C	*Stenotrophomonas maltophilia*	*M. sativa*	Bárdenas Reales	+++	++	++	-	+	-	-
**28**	MS O1-A	*Paenibacillus polymyxa*	*M. sativa*	Orce	-	+++	++	-	+	++	-
**29**	MS O1-B	*Paenibacillus polymyxa*	*M. sativa*	Orce	++	++	+++	-	+	++	-
**30**	MS M1-A	*Bacillus cereus*	*M. sativa*	Medinaceli	++	+	+++	+	+	+++	+
**31**	MS M1-B	*Stenotrophomonas maltophilia*	*M. sativa*	Medinaceli	++	+	++	+	+	-	+
**32**	MS M1-C	*Paenibacillus xylanexedens*	*M. sativa*	Medinaceli	++	+	+++	+	+	+	+
**33**	MS M1-D	*Enterobacter hormaechei*	*M. sativa*	Medinaceli	++	++	+	+	+	-	+
**34**	BB B2-A	*Paenibacillus amylolyticus*	*B. bituminosa*	Baza	++	++	++	-	+	++	+
**35**	BB B2-B	*Pantoea agglomerans*	*B. bituminosa*	Baza	++	+	+	-	+	+	+
**36**	BB B2-C	*Enterobacter hormaechei*	*B. bituminosa*	Baza	+++	+	+	+	+	-	+
**37**	BB B2-D	*Stenotrophomonas maltophilia*	*B. bituminosa*	Baza	++	+	+	-	+	-	+
**38**	MS VR1-A	*Stenotrophomonas maltophilia*	*M. sativa*	Vélez-Rubio	-	+++	+	-	+	++	-
**39**	MS VR1-B	*Stenotrophomonas maltophilia*	*M. sativa*	Vélez-Rubio	++	+	+	-	+	-	+
**40**	MS VR1-C	*Kosakonia cowanii*	*M. sativa*	Vélez-Rubio	+	+	++	-	-	-	+
**41**	MS Oeiras	*Paenibacillus polymyxa*	*M. sativa*	Oeiras	-	++	++	-	+	+++	+
**42**	BB Ni-A	*Stenotrophomonas maltophilia*	*B. bituminosa*	Níjar	++	+	++	+	+	-	+
**43**	BB Ni-B	*Paenibacillus tundrae*	*B. bituminosa*	Níjar	+	+	++	-	+	+++	+
**44**	BB Ni-C	*Erwinia* sp.	*B. bituminosa*	Níjar	++	++	+	+	+	-	+
**45**	BB Oeiras A	*Paenibacillus pabuli*	*B. bituminosa*	Oeiras	+++	+++	++	+	-	+++	+
**46**	BB VR-A	*Stenotrophomonas maltophilia*	*B. bituminosa*	Vélez-Rubio	+	++	+	-	+	-	+
**47**	BB VR-B	*Stenotrophomonas maltophilia*	*B. bituminosa*	Vélez-Rubio	++	++	+	+	+	-	+
**48**	BB VR-C	*Stenotrophomonas maltophilia*	*B. bituminosa*	Vélez-Rubio	++	++	++	+	+	-	-
**49**	BB Oeiras B	*Stenotrophomonas maltophilia*	*B. bituminosa*	Oeiras	-	+++	+	-	+	-	-
**50**	BB HO A	Unidentified	*B. bituminosa*	Huércal-Overa	+	+	-	+	-	-	-
**51**	BB HO B	Unidentified	*B. bituminosa*	Huércal-Overa	+	+	+	-	+	-	-

## Data Availability

Not applicable.

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
