# Peer review of "Geographically Disperse, Culturable Seed-Associated Microbiota in Forage Plants of Alfalfa (Medicago sativa L.) and Pitch Clover (Bituminaria bituminosa L.): Characterization of Beneficial Inherited Strains as Plant Stress-Tolerance Enhancers"

_biology, 2022, doi:10.3390/biology11121838_

Round 1

Reviewer 1 Report

I find the research work very interesting, and well written with a clear presentation of results and discussion.

I have a query regarding the pie chart drawn with the software. How reliable the software to draw the pie chart, is there any citation for it.

I tried my data with http://bioinformatics.psb.ugent.be/cgi-bin/liste/Venn/calculate_venn.htpl tool 

but failed to get the convincing data

Author Response

Dear editors and reviewers,

Firstly, thank you very much for the work and the comments. We really appreciate the interest and every suggestion that we believe will substantially improve the quality of this work. Below we present the changes we have made to it based on the comments and suggestions received:

Reviewer 1

I find the research work very interesting, and well written with a clear presentation of results and discussion.

I have a query regarding the pie chart drawn with the software. How reliable the software to draw the pie chart, is there any citation for it.

A: Thanks for the question. We have prepared the pie charts (I guess your question is about Figure 4) with Excel 2019. We used the average of the populations for each type of seed (M. sativa or B. bituminosa). It’s a simple set of data, so we consider the software and outputs are correct for the information we want to transmit. Hope this is enough as answer. Thanks again!

Some references to back the use of this software for this kind of graphs:

Raimund Schnürer, Martin Ritzi, Arzu Çöltekin, René Sieber. (2020) An empirical evaluation of three-dimensional pie charts with individually extruded sectors in a geovisualization context. Information Visualization 19:3, pages 183-206.

Iyad Abu Doush, Enrico Pontelli, Dominic Simon, Tran Cao Son, and Ou Ma. 2009. Making Microsoft Excel™: multimodal presentation of charts. In Proceedings of the 11th international ACM SIGACCESS conference on Computers and accessibility (Assets '09). Association for Computing Machinery, New York, NY, USA, 147–154. https://doi.org/10.1145/1639642.1639669

I tried my data with http://bioinformatics.psb.ugent.be/cgi-bin/liste/Venn/calculate_venn.htpl tool but failed to get the convincing data

A: Thanks for the comment. This online tool has been referenced before in our research articles (Vilchez et al. 2020 in Nature Plants), and it’s as well referenced by other authors (Božič et al. 2022 in Brain-Oxford Academic). The link you included in the comment I think is not correct (it is http://bioinformatics.psb.ugent.be/webtools/Venn/), but maybe you send me your output and I cannot open this website. I know the input boxes and format could be a little tricky, but the results are reliable. Hope this is enough to answer your doubt. Thanks!

Reviewer 2 Report

Dear Authors,

I attached my review in a separate file.

Kind regards

Author Response

Dear editors and reviewers,

Firstly, thank you very much for the work and the comments. We really appreciate the interest and every suggestion that we believe will substantially improve the quality of this work. Below we present the changes we have made to it based on the comments and suggestions received:

Reviewer 2

The review of a manuscript titled “Geographically disperse, cultivable seed-associated microbiota in forage plants of alfalfa (Medicago sativa L.) and pitch clover (Bituminaria bituminosa (L.) C.H.Stirt.): characterization of potential biotreatments based on beneficial inherited populations”. In the manuscript the Authors have made a great effort and a huge work on very important subject of beneficial bacterial populations.

In general, the paper is scientifically sound and the subject investigated is important for both the science and the agriculture. However, I have some remarks regarding to the clarity of description of the materials and methods, and the results sections. I am recommending it for publication, but after a major revision. Below I have presented all my remarks, small and big ones.

Line 15 Delete a word „even” because it is repeated twice in one sentence.

A: Really appreciate this comment! We have deleted in the text.

Line 70: why 80% alcohol, when usually the 70% dilution is said to have the most disinfectant efficiency?.

A: Really appreciate. In fact you’re right, it’s our mistake. Sometimes we used 75% to enhance the sterilization in complex samples, but this is not the case. This was just a typing mistake. So sorry.

Line 152-153: Why cfu per milligram of root dry weight?. An why not simply cfu/ml?. Now it is hard to refer, because the cfu/ml is a commonly used measurement of bacterial cells concentration. For instance, I don`t know if the number of cfu/mg of root dry weight is a high concentration of bacteria or not. Is it possible to add the concentration in cfu/ml?. If it is still possible, please do it.

A: We understand the question. The problem is, even if we try, is very difficult to control the initial number of samples we used to calculate the CFUs. If we want to normalize the results to compare each other fairly, we need to go to mg of biomass. Here, as not all the samples can be prepared in parallel in a reasonable time frame, and, moreover, different tissues can perform difference in water retention, fresh weight is no a good parameter to refer. We have noticed high internal variability when using fresh weight due to differential water lose (only the water loses in the sample caused a variation that is not registered once dry weight of the sample was used to normalize), so we decided to refer our counting to dry weight.

Some other areas suffer from similar problems and found in this way to normalize in a more reliable way. Please check Jeon et al. 2021, de Queiroz et al. 2016, Juhnke et al. 1989, Schreiter et al. 2018, or Ganesan et al. 2010.

We understand than maybe the cfu/ml is more comparable with other studies that initial sample is liquid, but is not completely correct when the original sample is solid and such variable. For example, our results have not meant just because each root set used for each replica have a different size, so the first dilution of each one is already massively different in cfu. In this case, more biomass is providing more CFUs, but even trying similar amount of roots, is very difficult control the quality of the data without normalize in terms of weight.

Finally, we have followed this normalization in previous publications after a careful scientific discussion with the scientific community (please see i.e., Vilchez et al., 2020 in Nature Plants). Hope this is good justification, we consider it is reasonable.

Lines 162-163: Please add the expected size of the PCR product.

A: Thanks for the suggestion. We included the expected size (about 700 bp) in the text.

Line 193: 96-well plate system. I have used many 96-well systems, but I don`t know the one that is describe in this. Please describe it more in more detailed way, because in this form it is insufficient for me to reproduce the test, and I can`t assess if it is suitable for the planned task.

A: Thanks for the suggestion, we have amplified this description in the text.

Line: 199: Why 0,45% NaCl instead of typical physiological solution – 0,9% NaCl?. Is there a purpose behind this?.

A: Thanks for the comment. To be honest, we were testing if some differences between 0.9 and 0.45% NaCl with a range of bacteria and we didn’t notice a difference, so we decided to minimize the use. We know is not an expensive reagent, but in our regular lab protocols sometimes we work with salt conditions and reducing the salinity of daily tasks helped us to control side effects. We have also published protocols and research article with this concentration before (please see Vilchez et al. 2020 in Nature Plants, Vilchez et al. 2022 in Bio-protocol, or Rodrigues-dos Santos et al. 2022 in Plants of MDPI)

Line 234: remove “then”. It is redundant.

A: Ok, good suggestion, Thanks! Now is delated in the text.

Subsection 2.5: Was an experiment control consisting of plants not treated with bacteria and grown in above 80% SRH and below 20% SRH included?. To assess the influence of potentially plant growth promoting bacteria alone and to assess the effect of the draught itself?. To be able to compare the groups?. If not, how did you compare the groups of plants? On what basis did you assess a plant growth-promoting activity of the tested bacteria?. Please explain. Why mock (figure 8, line 692) is not described in the material and methods section? Is mock a control consisted of plants not treated with bacteria. If yes, please describe it more precisely.

A: We really appreciate this comment. As you suggested we included some modifications in the text to clarify. Hope now is solved. Thanks!

Culturable bacteria:

In Materials and Methods section (line 139) you write that you have analysed a 60-seed sets, as I understand, the seeds were collected form 18 locations (“in 9 locations for each type of plant” line 115). So I assume that 19 samples of 60-seed sets were analysed for germination process, and that the culturable endophytic bacteria were isolated from the root of plants that germinated from each of that 19 samples=locations. And so, as a result, I would expect a 19 sets of bacteria identifications – one set of bacterial species for each sampling spot, because only than one could tell if geographic location has any impact on the occurrence of plant growth promoting bacteria.

A: Thanks for your comment! I understand this part was complicated (at least to write, seems also to read), so sorry. The 60-seed sets are for each location (added to the text to clarify), and thereafter in Figure 5 and Table 3, they are described. Sorry again, this was difficult to build so some info is far away from the description, but we didn’t find a better way to compose everything better. Sorry again.

Instead the Authors write that “we were able to isolate a total of 51 strains …” (line 375). Perhaps I have missed something but please explain how can you tell which strain is from what localization?. Did you pulled the seeds from all locations to isolate the bacteria out of them?. Please explain. I understand that Figure 5 should be the answer to my remark, however the methods and results should be described in a concise manner. I read the manuscript thoroughly and only after I saw the Figure 5 (line 480 – much further in the text), I was confused about the description. Please rewrite it. I still don’t know where did those “51 isolates” came from?. How the number 51 is related to the number of strains listed in Figure 5?. You say that “In the collection of 51 strains isolated, a total of 16 unique species were identified” (line 410), so it is not that. Than what?. Please explain.

A: Thanks for this comment, we find it very useful to avoid confusions. We have included some tables to explain where each strain was isolated from. We have included some changes in the texts to help the general meaning. Really appreciate!

Lines 418-422: If the isolates had “not enough pair based amplified to ensure the identification” (lines 419-420) than they could not be compared in the phylogenetic analysis because such analysis require a set of DNA sequences aligned with each other and of the same length. You cannot compare long and short sequence in one assay. The results obtained in this way are not correct, as they could be both false-positive and false-negative. Please correct that section including Figure 4c. Additionally nowhere in the manuscript I found the information about the GenBank accession numbers of the sequences that you obtained. It must be provided. This is a standard practice.

A: Thanks for the suggestion! We are aware of this; however, we have not included in a correct description of how we did it. Now is changed in the text. In brief, we performed the phylogenetical analysis only with those sequences with quality and bases enough (correspondent to the expected length), but thereafter once they were correctly aligned, we include the sequence with not enough bp to find what was the closest related sequences. This was performed not with the aim of identify them, but to place them in context. 

Sorry, we forget to include the accession number, really appreciate you realize! Some of the sequences are still being processed, but they should be fully available by 12th December (as GenBank notified us). Now is included in text, Thanks again!

Lines 422-433: “Considering the geographical location, the North isolation zone ('Castejón de Monegros', 'Belchite', 'Bárdenas Reales' and 'Medinaceli') provided a total of 13 strains, identifying 6 different species (A. hermannii, E. hormaechei, S. maltophilia , B. cereus, P. polymyxa and P. xylanexedens); - So you did analyse the occurrence of bacteria separately for each location tested. That kind of data set I meant when I was asking about bacterial identification, two sections above. Please describe it in a consistent way.

A: Thanks for the comment, sorry for the misunderstanding. We are included some changes in the text to help with this.

Lines 435-437: I quote: “In this way, most of the strains (43) were isolated from plant seed in soils with a pH ˃ 9, EC below 4 dS/m (31 strains) and a soil water content below 0.75% 436 (35 strains).” – How does that refer to “we were able to isolate a total of 51 strains …” (line 375)?. Please explain or rewrite the manuscript.

A: Thanks for the comment. They are not summing up values, it refers to which number of the strains were isolated from each environment. However, we agreed that written in this way could be confusing, so we rearrange these concepts. Thanks!

Figure 7 (line 606): What do you mean by “screening skills” (line 606)?. Perhaps you meant a “Comparison of the biochemical properties among isolated strains”?. Or something similar?. Please explain

A: Thanks for the comment. Here we mean that those biochemical tests are used for a screening to select the candidates to use as bioinoculant. This graph is the only place all tests are reflected together. However, the reading could be confusing, as you noticed, so we have included a change in the text. Thanks!

Also, within the text please decide whether you want to use the term “strain” or “isolate” and be consistent about it, because this is not the same.

A: Thanks at all. You’re completely right, we have checked all text to apply this change.

Line 547: Remove “here”. It is redundant.

A: Thanks for the comment, now is removed.

The Conclusions section: In my opinion this is mere summary of the result. The closest to conclusion, to me, is the notion “Different strains of Stenotrophomonas maltophilia and Paenibacillus polymyxa were isolated in both model plants despite the diverse origins and environmental context, suggesting a conserved or common inherited microbiota” (lines: 862- 864).

A: Thanks for the comment. We have modified this section in accordance.  

Reviewer 3 Report

Dear Authors,

                I reviewed the manuscript titled "Geographically disperse, cultivable seed-associated microbiota in forage plants of alfalfa (Medicago sativa L.) and pitch clover (Bituminaria bituminosa (L.) C.H.Stirt.): characterization of po- tential biotreatments based on beneficial inherited populations" and found very intersting. Overall the study is designed well and organized in a good way and should be accepted for publication after minor revision. 

In summary, while the objectives of the research work are clear and experimental approach appropriate, the MS in its current form some revision.

Although the title is more explanatory but very lenghthy and need to be shortened.

The use of english language is poor and some sentences are not easily understandable and therefore need to be improved. 

Author Response

Reviewer 3

I reviewed the manuscript titled "Geographically disperse, cultivable seed-associated microbiota in forage plants of alfalfa (Medicago sativa L.) and pitch clover (Bituminaria bituminosa (L.) C.H.Stirt.): characterization of potential biotreatments based on beneficial inherited populations" and found very interesting. Overall, the study is designed well and organized in a good way and should be accepted for publication after minor revision. 

In summary, while the objectives of the research work are clear and experimental approach appropriate, the MS in its current form some revision.

Although the title is more explanatory but very lengthy and need to be shortened.

A: Thanks for the suggestion. We have slightly modified the title to make it shorter. However, we consider it necessary to keep most of the wording to ensure its visibility and different approaches to find it later as a referential work.

Dear editors and reviewers,

Firstly, thank you very much for the work and the comments. We really appreciate the interest and every suggestion that we believe will substantially improve the quality of this work. Below we present the changes we have made to it based on the comments and suggestions received:

The use of English language is poor, and some sentences are not easily understandable and therefore need to be improved. 

 A:  We really appreciate the suggestion. We have made a full revision of the use of English and consult an external colleague (native) to make some more corrections. Hope you can find the text well now. Thanks for the suggestion!

Round 2

Reviewer 2 Report

Dear Authors,

Thank you very much for the answers to my remarks. I accept all of them. 

The goal of my remarks was to increase the clarity of the experiment description and increase the overall merit of the whole manuscript. Judging by your answers you agreed with me. 

I have found some spelling mistakes, and I have two small suggestions:

Line 217 Misspelling - is "Orther" instead of "order"

Line 400 Misspelling - is "lager" should be "larger"

Line 901 I would avoid words like "remarkably" in the scientific paper. Please remove this word.

Figure 4c I like when the accession numbers are included in the name of the analyzed strains in a phylogenetic tree. It simplifies both the presentation and the reception of the data. But it is not mandatory, and I leave the decision to the Authors. 

 To sum up, I recommend the manuscript for publication, after the correction of minor spelling mistakes (listed above). 

Kind regards,

The Reviewer

Author Response

Dear reviewer and editors,

We really appreciate the comments and suggestions, they really helped a lot! In fact, these last ones are very important to avoid shaming wording misspelling, thanks again! We have applied these last changes as suggested. About these comments:

- Line 901 I would avoid words like "remarkably" in the scientific paper. Please remove this word.

A: We have to change most of them, specially where they could make the text seems less objective by using them. We just kept in the occasions that make it relevant in the text.

- Figure 4c I like when the accession numbers are included in the name of the analyzed strains in a phylogenetic tree. It simplifies both the presentation and the reception of the data. But it is not mandatory, and I leave the decision to the Authors. 

A: We fully understand, we also like in the other way, but after trying, the figure get very crowded and even more difficult to read, so we think to leave in the current shape is better for an easier interpretation and cleaner reading. Hope this is ok.

Thanks again for the effort and the help, it is really valuable for us.

Best,

Dr. Vilchez